# SCALABLE TRAINING FOR VECTOR-QUANTIZED NETWORKS WITH 100% CODEBOOK UTILIZATION

**Yifan Chang**[1,4]    **Jie Qin**[2]    **Limeng Qiao**[2]    **Xiaofeng Wang**[3]

**Zheng Zhu**[3]    **Lin Ma**[2]    **Xingang Wang**[1,5]

[1]CASIA    [2]Meituan    [3]GigaAI    [4]UCAS

[5]Luoyang Institute for Robot and Intelligent Equipment

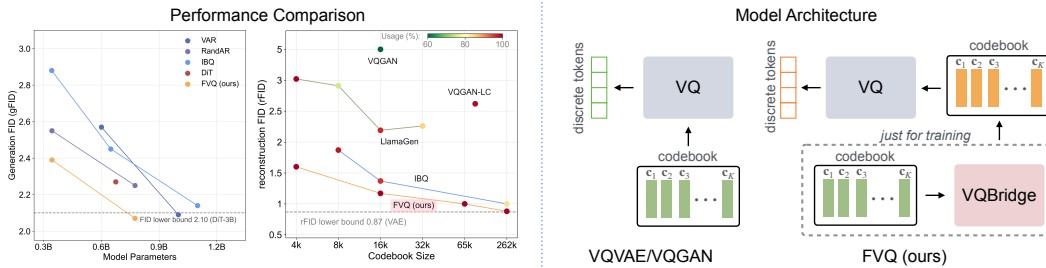

Figure 1: **Performance comparison and model architecture**. **Left**: Performance comparison showing generation (FID) and reconstruction (rFID) quality across different methods. **Right**: Architectural comparison between baseline VQVAE/VQGAN and the proposed FVQ model, with VQBridge component for training.

## ABSTRACT

Vector quantization (VQ) is a key component in discrete tokenizers for image generation, but its training is often unstable due to straight-through estimation bias, one-step-behind updates, and sparse codebook gradients, which lead to suboptimal reconstruction performance and low codebook usage. In this work, we analyze these fundamental challenges and provide a simple yet effective solution. To maintain high codebook usage in VQ networks (VQN) during learning annealing and codebook size expansion, we propose VQBridge, a robust, scalable, and efficient projector based on the map function method. VQBridge optimizes code vectors through a compress–process–recover pipeline, enabling stable and effective codebook training. By combining VQBridge with learning annealing, our VQN achieves full (100%) codebook usage across diverse codebook configurations, which we refer to as FVQ (FullVQ). Through extensive experiments, we demonstrate that FVQ is effective, scalable, and generalizable: it attains 100% codebook usage even with a 262k-codebook, achieves state-of-the-art reconstruction performance, consistently improves with larger codebooks, higher vector channels, or longer training, and remains effective across different VQ variants. Moreover, when integrated with LlamaGen, FVQ significantly enhances image generation performance, surpassing visual autoregressive models (VAR) by 0.5 and diffusion models (DiT) by 0.2 rFID, highlighting the importance of high-quality tokenizers for strong autoregressive image generation. https://github.com/yfChang-cv/FVQ

## 1 INTRODUCTION

The scalability of autoregressive models has been well demonstrated in large language models (Radford et al., 2018; Brown et al., 2020; Touvron et al., 2023; Grattafiori et al., 2024; Bai et al., 2023). Inspired by this, pioneering works (Esser et al., 2021; Yu et al., 2022; Ramesh et al., 2021; Tian et al., 2024) have explored the application of autoregressive models in visual generation tasks. To enhance scalability and reduce computational costs, a common approach (Van Den Oord et al., 2017; Razavi et al., 2019) involves introducing discrete tokenizers that employ vector quantization (VQ) to convert continuous images into discrete tokens. Autoregressive models are trained on these tokens in a self-supervised strategy to predict the next tokens. Generally, the reconstruction capability of the tokenizer directly determines the upper bound of the generative model's performance (Esser

et al., 2021; Yu et al., 2024a). Therefore, the design of discrete tokenizers plays a crucial role in autoregressive visual generation.

However, due to the information loss introduced by vector quantization, the performance of discrete tokenizers has long been limited compared with continuous tokenizers (Kingma, 2013). The intuitive insight is that enlarging the codebook could reduce the information loss during quantization. Nevertheless, prior work (Yu et al., 2022; Sun et al., 2024; Ma et al., 2025) has shown that simply increasing the codebook size or code dimension often leads to codebook collapse, which causes a severe reduction in codebook usage and consequently undermines the generative capability of downstream models (Huh et al., 2023; Luo et al., 2024). Despite many efforts to mitigate this issue, when scaling to larger settings (*e.g.*, codebook size of 262k and code dimension of 256) (Shi et al., 2025), full codebook usage remains unattainable and the performance is still unsatisfactory.

To address this issue, we revisit VQ from fundamental mechanism and three fundamental challenges (Huh et al., 2023), together with empirical observations that shed light on resolving them. VQ relies on the straight-through estimator (STE) (Bengio et al., 2013) to resolve the non-differentiability problem, but this introduces three additional challenges: straight-through estimation bias, one-step-behind update, and sparse codebook gradients. The first two challenges cause misalignment in optimization across the network, leading to instability, while sparse codebook gradients prevent the codebook from being sufficiently trained. In practice, only a small subset of code vectors are updated, which is the primary cause of collapse.

Surprisingly, based on our observations, these challenges can be effectively mitigated using only learning annealing and a sufficient powerful projector that jointly optimizes the codebook. Prior works (Zhu et al., 2024a;b) employ a linear layer as the projector for joint codebook optimization, showing some improvement. However, our further experiments reveal that a linear projector alone is fragile: it makes the network highly sensitive to the learning rate and insufficiently capable when scaling to larger codebooks. To enable robust and scalable VQ training, we propose a robust, scalable and efficient projector, called VQBridge, as shown in Figure 1. VQBridge adopts a compress–process–recover pipeline: it first compresses the set of code vectors into a smaller number, then models interactions via ViT blocks (Dosovitskiy et al., 2020), and finally recovers the vectors to the original codebook size and dimensionality. This design allows VQ to rapidly achieve and sustain 100% codebook usage for arbitrary codebook configurations, enabling effective optimization of VQ.

Using VQBridge combined with learning annealing, we conduct extensive experiments to validate its effectiveness, scalability, and generalization. On the ImageNet benchmark (Deng et al., 2009), FVQ consistently achieves 100% codebook usage across all configurations, including a 262k-codebook, while substantially enhancing the reconstruction performance of discrete tokenizers (rFID = 0.88), without introducing additional parameters or inference cost. FVQ exhibits strong scalability, with performance consistently improving as the codebook size, vector channel, and number of training epochs increase. Furthermore, the approach remains effective when applied to multi-code representation VQ (Lee et al., 2022; Tian et al., 2024), demonstrating robust generalization. Besides, FVQ leads to substantial improvements in generation performance: for instance, LlamaGen-XL improves FID from 3.39 to 2.07, outperforming comparable visual autoregressive models VAR (Tian et al., 2024) (FID=2.57) and diffusion models DiT (Peebles & Xie, 2023) (FID=2.27). These results highlight that even a vanilla autoregressive framework can achieve strong generation capabilities when the tokenizer exhibits high-quality reconstruction.

In summary, our contributions include:

1. We analyze the fundamental challenges arising from vector quantization and provide effective observations, leading to a simple yet scalable training framework (FVQ).

2. Through extensive experiments, we show that FVQ is effective, scalable, and generalizable. Specifically, it achieves 100% codebook usage across all configurations, attains state-of-the-art reconstruction performance, and consistently improves as the codebook size, vector channel, and number of training epochs increase, remains effective across different types of VQ.

3. We demonstrate that FVQ substantially enhances the generative capability of autoregressive models. Without any bells and whistles, a vanilla autoregressive framework equipped with FVQ outperforms advanced visual autoregressive models (VAR) and diffusion models (DiT).

## 2 BACKGROUND

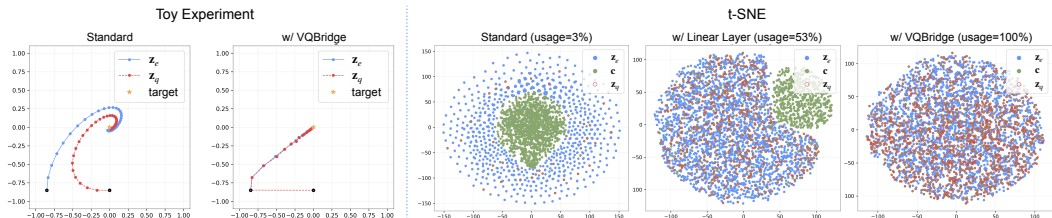

Figure 2: **Toy experiment and t-SNE visualization results. Left**: A dynamic vector quantization toy experiment where the target is a fixed optimization objective and the model is optimized with SGD. Using the standard method with only STE leads to a spiral trajectory, while VQBridge produces a piecewise linear trajectory. With VQBridge, $\mathbf{z}_q$ closely follows $\mathbf{z}_e$, reducing their distance already in the early stages of training. **Right**: Due to sparse gradients from the codebook, the standard method achieves only 3% codebook usage. Adding a linear layer for joint optimization improves the usage but remains insufficient, whereas VQBridge achieves 100% usage.

**Vector Quantized Network.** A vector quantized network (VQN) contains Encoder $\mathcal{E}(\cdot)$, Vector Quantization Layer $\mathcal{Q}(\cdot, \cdot)$ and Decoder $\mathcal{D}(\cdot)$. Given a vector $\mathbf{x}$, the encoder maps it into a latent representation $\mathbf{z}_e$. The codebook is a collection of $K$ code vectors, denoted as $\mathcal{C} = \{\mathbf{c}_1, \mathbf{c}_2, \ldots, \mathbf{c}_K\}$, where each vector $\mathbf{c}_i$ has same dimension as $\mathbf{z}_e$. In quantization layer $\mathcal{Q}(\cdot, \cdot)$, $\mathbf{z}_e$ is quantized into $\mathbf{z}_q$ by assigning it into a code vector from codebook $\mathcal{C}$ using a distance measure $d(\cdot, \cdot)$:

$$\mathbf{z}_q = \mathbf{c}_x, \quad \text{where} \quad x = \arg\min_i d(\mathbf{z}_e, \mathbf{c}_i). \tag{1}$$

Therefore, VQNs can be denoted:

$$\hat{\mathbf{y}} = \mathcal{D}(\mathcal{Q}(\mathcal{E}(\mathbf{x}), \mathcal{C})) = \mathcal{D}(\mathcal{Q}(\mathbf{z}_e, \mathcal{C})) = \mathcal{D}(\mathbf{z}_q). \tag{2}$$

VQNs predict the output $\hat{\mathbf{y}}$ and compute loss with target $\mathbf{y} : \mathcal{L}(\mathbf{y}, \hat{\mathbf{y}})$. The objective of VQNs is to optimize the following loss function:

$$\min_{\mathcal{E}, \mathcal{Q}, \mathcal{D}} \mathbb{E}_{(\mathbf{x}, \mathbf{y}) \sim \mathcal{T}}[\mathcal{L}_{task}(\mathcal{D}(\mathcal{Q}(\mathcal{E}(\mathbf{x}), \mathcal{C}))), \mathbf{y})], \tag{3}$$

where $\mathcal{T}$ is training dataset. The extension to image reconstruction is straightforward by applying the formulation to 2D data; see Appendix F for details.

**Optimization Challenge.** Since the $\arg\min$ operation in the quantization layer $\mathcal{Q}(\cdot)$ is non-differentiable, a common approach is to apply the straight-through estimator (STE) during backpropagation (Bengio et al., 2013), which allows gradients to bypass the discrete operation:

$$\mathbf{z}_q = \mathbf{z}_e + sg[(\mathbf{z}_q - \mathbf{z}_e)], \tag{4}$$

where $sg[\cdot]$ is stop-gradient operation. In addition, a commitment loss (Van Den Oord et al., 2017) is introduced to ensure that the gradient flow on both sides of the quantization layer remains accurate:

$$\mathcal{L}_{cmt}(\mathbf{z}_e, \mathbf{z}_q) = d(\mathbf{z}_q, sg[\mathbf{z}_e]) + \beta d(\mathbf{z}_e, sg[\mathbf{z}_q]). \tag{5}$$

Although this approach makes the objective differentiable and thus optimizable, it introduces several additional challenges.

*Challenge 1:* Straight-Through Estimation Bias. The straight-through estimation error, defined as $\delta = \mathbf{z}_q - \mathbf{z}_e$, introduces bias into the training process. It directly affects the decoder $\mathcal{D}(\cdot)$ and indirectly impacts the encoder $\mathcal{E}(\cdot)$ through the commitment loss (Huh et al., 2023).

*Challenge 2:* One-step-behind Update. Through derivation (see Appendix D), we show that the commitment loss causes the model to align with historical representations instead of the current ones (Łańcucki et al., 2020; Huh et al., 2023). Specifically, the codebook is updated as

$$\mathbf{z}_q^{(t+1)} \leftarrow (1 - \eta) \cdot \mathbf{z}_q^{(t)} + \eta \cdot \mathbf{z}_e^{(t)}, \tag{6}$$

and the decoder is trained on the previous codebook output. This mismatch leads to a misalignment between the encoder and decoder. Ideally, both should be updated using the current representation:

$$\mathbf{z}_q^{(t+1)} \leftarrow (1 - \eta) \cdot \mathbf{z}_q^{(t)} + \eta \cdot \mathbf{z}_e^{(t+1)}. \qquad (7)$$

***Challenge 3:*** Sparse Codebook Gradients. The codebook $\mathcal{C}$ suffers from sparse gradients—only the selected code vectors receive updates, while the unselected ones get no gradient signal and are likely to remain unused indefinitely (Zheng & Vedaldi, 2023; Shi et al., 2025). This results in codebook collapse and limited representational capacity.

We use a toy experiment to illustrate the impact of the STE. As shown in Figure 2, the optimization trajectories of $\mathbf{z}_e$ and $\mathbf{z}_q$ form a spiral shape, revealing hindered optimization (**Challenge 1 & 2**). Moreover, the t-SNE visualization further shows that directly applying the STE leads to low usage, with most codebook entries left unused. This phenomenon arises from the sparse gradients received by the codebook (**Challenge 3**).

## 3 METHODOLOGY

### 3.1 PRELIMINARY OBSERVATIONS

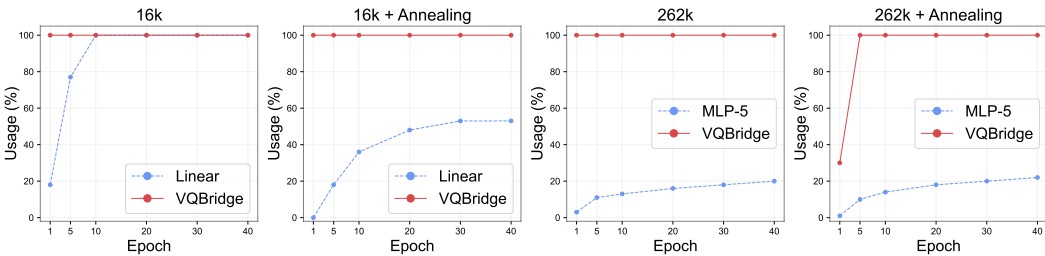

Figure 3: **Codebook usage during training.** With a codebook size of 16k, the linear-layer baseline can achieve 100% usage, but usage drops significantly when learning annealing is applied. With a larger codebook size of 262k, even a 5-layer MLP fails to maintain sufficient usage. In contrast, VQBridge rapidly achieves 100% usage under all configurations and with learning annealing, which greatly facilitates VQN optimization.

We denote the corresponding sets of $\mathbf{z}_e$, $\mathbf{c}$, and $\mathbf{z}_q$ as $\mathcal{P}_z$, $\mathcal{C}_z$, and $\mathcal{H}_z$, respectively. The distance between the distributions of the two sets can be formally defined as

$$D(\mathcal{P}_z, \mathcal{C}_z) = \frac{1}{|\mathcal{P}_z|} \sum_{\mathbf{z}_i \in \mathcal{P}_z} \min_{\mathbf{c}_i \in \mathcal{C}_z} d(\mathbf{z}_i, \mathbf{c}_i).$$

Our work is based on the simple and effective linear reparameterization (Huh et al., 2023; Zhu et al., 2024a;b), which can be generalized as a mapping function $f(\cdot)$ from $\mathcal{C}_z$ to a new set $\hat{\mathcal{C}}_z$, with joint optimization over $\mathbf{c}$: $\hat{\mathcal{C}}_z = f(\mathcal{C}_z)$. Consequently, the training objective shifts to minimizing the distance between $\mathcal{P}_z$ and $\hat{\mathcal{C}}_z$: $D(\mathcal{P}_z, \hat{\mathcal{C}}_z)$. A smaller $D(\mathcal{P}_z, \mathcal{C}_z)$ indicates a smaller estimation error $\delta$. However, estimation error does not necessarily reflect optimization effectiveness, since a small number of active codebook vectors may lead to lower loss but fail to yield meaningful performance gains (Huh et al., 2023). Codebook usage can therefore serve as a complementary metric, as higher usage indicates that more vectors are effectively contributing, which is more informative.

**Observation 1** *To mitigate the straight-through estimation bias, rapid alignment of the two distributions is crucial, reducing $D(\mathcal{P}_z, \mathcal{C}_z)$ early in training. Furthermore, estimation error must remain low for stable optimization. In practice, this requires using a larger codebook while maintaining high usage from early to late training. We present the effect of codebook size on the loss in Appendix C.*

**Observation 2** *Learning annealing is necessary, as a smaller learning rate helps reduce the one-step-behind effect and improves the overall alignment of the model. We establish an upper bound on the effect of the one-step-behind update and demonstrate that learning annealing serves as an effective remedy. Further derivations are presented in Appendix E.*

**Observation 3** *Linear layer is weak. As shown in Figure 3, we observe that when a linear layer is used as the mapping function $f(\cdot)$, the codebook usage drops significantly during learning annealing. Additionally, under codebook scaling, the linear mapping struggles even more. Attempts to increase model capacity with a 5-layer MLP still result in poor usage. This suggests that the representational power of such mappings is inadequate for aligning the distributions of $\mathcal{P}_z$ and $\hat{\mathcal{C}}_z$.*

Based on the above observations, stable and scalable training of VQN requires a strong and robust mapping function $f(\cdot)$, combined with learning annealing. In the following section, we introduce our proposed design, referred to as VQBridge.

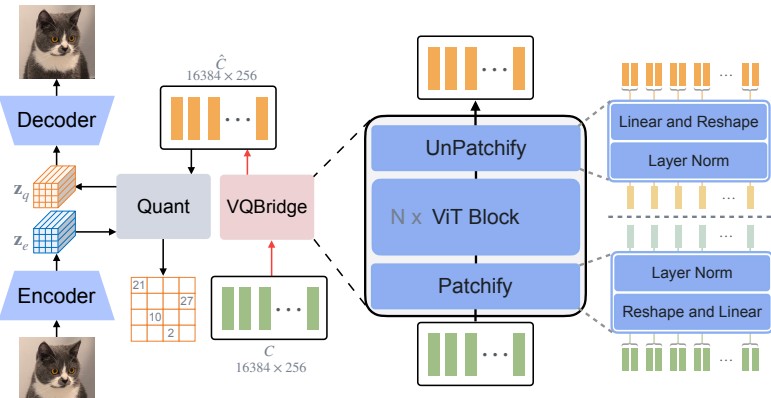

Figure 4: **FVQ framwork overview.** VQBridge first applies 1D patchification to the code vectors, then performs global interaction using ViT blocks, and finally restores them to their original size.

## 3.2 VQBRIDGE DESIGN

The design objective of VQBridge is to achieve strong fitting capability, enabling rapid alignment between two distributions, while maintaining scalability and training stability. Inspired by the architecture of Diffusion Transformers (DiT) (Peebles & Xie, 2023), the module follows a compress–process–recover pipeline, where the compression step enables efficient scaling to large codebooks (*e.g.*, 262k entries). To simplify the problem, we assume that codebook $\mathcal{C}$ and the mapped codebook $\hat{\mathcal{C}} = f(\mathcal{C})$ share the same size and vector channel. Details are provided in Figure 4.

First, we perform a 1D *patchify* operation. Given a codebook $\mathcal{C}$ containing $K$ code vectors, we divide it into $p$ groups, each containing $\frac{K}{p}$ vectors. For each group, the vectors are compressed into a single vector via a shared linear projection $W_{\text{comp}} \in \mathbb{R}^{\frac{p}{Kd} \times d'}$, followed by a LayerNorm operation:

$$\mathbf{h}_g = \text{LN}\big(\mathbf{C}_g W_{\text{comp}}\big), \quad g = 1, \dots, p. \tag{8}$$

Next, $N$ ViT blocks are applied as scalable layers to process the intermediate representation $\mathbf{H} \in \mathbb{R}^{p \times d'}$, $\mathbf{H} = [\mathbf{h}_1, \mathbf{h}_2, ..., \mathbf{h}_p]$, where each $\mathbf{h}_g$ is a compressed intermediate representation. This design enables global information exchange and allowing the model to fit complex relationships between code vectors:

$$\mathbf{H}' = \text{ViT}^N(\mathbf{H}). \tag{9}$$

Finally, a 1D *unpatchify* operation restores the representation to its original size. Specifically, $\mathbf{H}'$ is first normalized via LayerNorm, then expanded to $\frac{Kd}{p}$ times its dimension through a shared linear projection $W_{\text{exp}} \in \mathbb{R}^{d' \times \frac{Kd}{p}}$:

$$\tilde{\mathbf{H}} = \text{LN}(\mathbf{H}')W_{\text{exp}}, \tag{10}$$

followed by a reshape operation that converts each expanded vector back into $p$ vectors within group:

$$\hat{\mathbf{C}} = \text{reshape}(\tilde{\mathbf{H}}) \in \mathbb{R}^{K \times d}. \tag{11}$$

This yields a codebook $\hat{\mathbf{C}}$ of the same size as the input to VQBridge, where the code vectors have undergone full mutual interaction.

Note that both VQBridge and the original codebook $\mathcal{C}$ can be discarded after training, maintaining only the mapped codebook $\hat{\mathcal{C}}$. This implies that VQBridge introduces no additional computational overhead during inference, preserving the original VQN's workflow and computational cost.

## 4 EXPERIMENTS

### 4.1 MAIN EXPERIMENTS

Table 1: **Reconstruction performance of different tokenizers on $256 \times 256$ ImageNet 50k validation set using the same Encoder and Decoder.** * trained on OpenImages (Kuznetsova et al., 2020). † trained for 330 epochs. ‡ trained for 120 epochs. We highlight the best results in bold and underline the second-best results.

| Method | Tokens | Codebook Size | Vector Channel | rFID↓ | LPIPS↓ | Codebook Usage↑ |
|---|---|---|---|---|---|---|
| VQGAN (Esser et al., 2021) | $16 \times 16$ | 1,024 | 256 | 7.94 | – | 44% |
| VQGAN (Esser et al., 2021) | $16 \times 16$ | 16,384 | 256 | 4.98 | 0.17 | 5.9% |
| SD* (Rombach et al., 2022a) | $16 \times 16$ | 16,384 | 8 | 5.15 | – | – |
| MaskGIT (Chang et al., 2022) | $16 \times 16$ | 1,024 | 256 | 2.28 | – | – |
| LlamaGen (Sun et al., 2024) | $16 \times 16$ | 16,384 | 8 | 2.19 | 0.14 | 97% |
| LlamaGen (Sun et al., 2024) | $16 \times 16$ | 16,384 | 256 | 9.21 | – | 0.29% |
| VQGAN-LC (Zhu et al., 2024a) | $16 \times 16$ | 16,384 | 8 | 3.01 | 0.13 | 99% |
| VQGAN-LC (Zhu et al., 2024a) | $16 \times 16$ | 100,000 | 8 | 2.62 | **0.12** | 99% |
| IBQ† (Shi et al., 2025) | $16 \times 16$ | 16,384 | 256 | 1.55 | 0.23 | 97% |
| FVQ | $16 \times 16$ | 16,384 | 256 | 1.30 | 0.13 | **100%** |
| FVQ‡ | $16 \times 16$ | 16,384 | 256 | **1.17** | 0.13 | **100%** |

Table 2: **Generative model comparison on class-conditional ImageNet $256 \times 256$.** The metrics include Fréchet Inception Distance (FID), Inception Score (IS), Precision (Pre), and Recall (Rec). We highlight the best results in bold and underline the second-best results.

| Type | Model | #Para | FID↓ | IS↑ | Pre↑ | Rec↑ |
|---|---|---|---|---|---|---|
| Diff. | DiT-L/2 (Peebles & Xie, 2023) | 458M | 5.02 | 167.2 | 0.75 | 0.57 |
| Diff. | DiT-XL/2 (Peebles & Xie, 2023) | 675M | 2.27 | 278.2 | 0.83 | 0.57 |
| Mask. | MaskGIT (Chang et al., 2022) | 227M | 6.18 | 182.1 | 0.80 | 0.51 |
| VAR | VAR-$d16$ (Tian et al., 2024) | 310M | 3.30 | 274.4 | **0.84** | 0.51 |
| VAR | VAR-$d20$ (Tian et al., 2024) | 600M | 2.57 | **302.6** | 0.83 | 0.56 |
| AR | VQGAN (Esser et al., 2021) | 227M | 18.65 | 80.4 | 0.78 | 0.26 |
| AR | LlamaGen-L (Sun et al., 2024) | 343M | 3.81 | 248.3 | 0.83 | 0.52 |
| AR | LlamaGen-XL (Sun et al., 2024) | 775M | 3.39 | 227.1 | 0.81 | 0.54 |
| AR | IBQ-B (Shi et al., 2025) | 342M | 2.88 | 254.4 | **0.84** | 0.51 |
| AR | IBQ-L (Shi et al., 2025) | 649M | 2.45 | 267.5 | 0.83 | 0.52 |
| AR | FVQ-L (LlamaGen-L) | 343M | 2.39 | 276.6 | **0.84** | 0.56 |
| AR | FVQ-XL (LlamaGen-XL) | 775M | **2.07** | 287.0 | 0.83 | **0.58** |

**Implementation Details.** For visual reconstruction, we adopt a codebook with a size of 16,384 and a vector channel of 256. VQBridge is configured with a latent dimension of 256, a depth of 2 and a patch size of 16. The encoder and decoder configurations are identical to those in VQGAN (Esser et al., 2021; Sun et al., 2024), utilizing 2 layers of ResBlocks. The model is trained for 40 epochs with a base learning rate of 1e-4 and a batch size of 128. Learning annealing starts with a 4-epoch warmup, followed by a learning rate decay to 1% of the original value. The Adam optimizer (Kingma, 2014) is employed with $\beta_1 = 0.9$ and $\beta_2 = 0.95$. Finally, we scale the training time to 120 epochs. For visual generation, LlamaGen (Sun et al., 2024) is utilized as the generator, with training configurations aligned with the open-source implementations. Additional details are provided in the Appendix G.

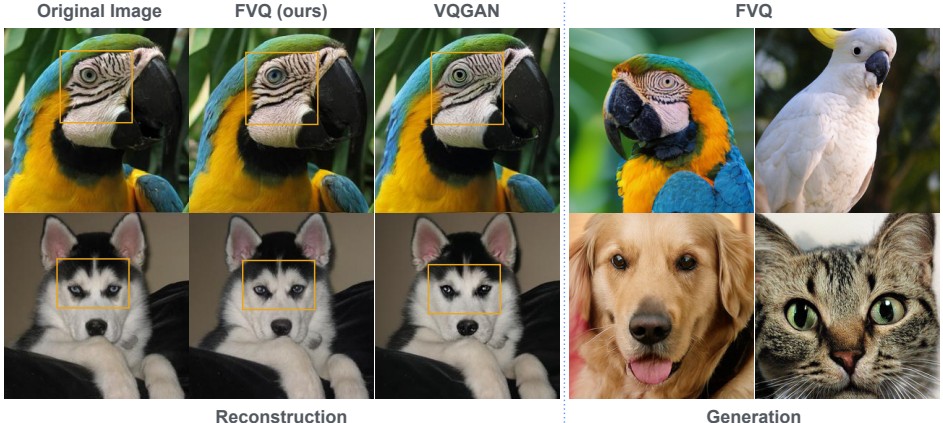

**Reconstruction**            **Generation**

Figure 5: **Qualitative results of FVQ. Left**: reconstruction performance compared against baseline methods. **Right**: generation results.

**Reconstruction Results.** In Table 1, we compare existing tokenizers using the same encoder and decoder architectures, most of which are trained on ImageNet-1k 256x256 with a downsampling ratio of 16. FVQ achieves an rFID of 1.30 after 40 epochs, outperforming all other discrete tokenizers. Scaling the training time to 120 epochs further improves the rFID to 1.17. By addressing the optimization challenges of vector quantization, our method can be extended to other discrete tokenizers, unlocking their full potential (see Section 4.4).

**Visual Generation.** We report the results of LlamaGen (Sun et al., 2024) as generators in Table 2, comparing them with prior generative models. FVQ improves the FID of LlamaGen-L from 3.81 to 2.39, surpassing other models with similar parameter sizes. Notably, FVQ-L (343M) even outperforms VAR-$d20$ (600M, 2.39 *vs.* 2.57), and FVQ-XL exceeds DiT-XL/2 (2.07 *vs.* 2.27), demonstrating the potential of autoregressive (AR) methods. These results further emphasize that the tokenizer is the key to AR visual generation. Previous AR models are constrained by suboptimal tokenizers, whereas FVQ significantly enhances the generative capabilities of AR models, fully unlocking their potential. See Figure 5 for visualization results.

## 4.2 ABLATION STUDIES

To identify the most efficient and effective setup, we explore various settings of VQBridge in Figure 6, focusing on the interplay between its parameters, codebook's size and vector channel.

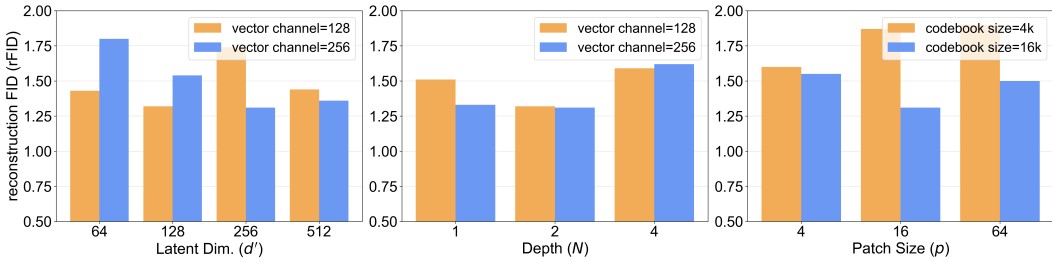

Figure 6: **VQBridge desin ablation results.** We vary VQBridge configurations including the latent dimension ($d'$), ViT block depth ($N$), and patch size ($p$), as well as codebook parameters such as the vector channel and codebook size.

**Latent Dimension** ($d'$). In left subfigure, increasing $d'$ from 64 to 256 consistently improves the reconstruction quality. Interestingly, the best results appear when the latent dimension $d'$ matches the vector channel size, indicating that a balanced latent embedding dimension and code vector width facilitates more effective representation learning. Further enlarging $d'$ to 512 does not provide additional benefits, and may even degrade performance due to over-parameterization and optimization difficulties. Therefore, we set $d'$ equal to the vector channel size in our final configuration.

**Depth of ViT Block** ($N$). The middle subfigure shows that using a single ViT block leads to suboptimal results. Increasing the depth to $N = 2$ significantly improves performance, while further stacking blocks ($N = 4$) actually degrades performance due to increased optimization difficulty and higher computation cost. Therefore, $N = 2$ is adopted as our default.

**Patch Size** ($p$). The right subfigure presents the effect of patch size under different codebook sizes, with both parameters following a $4^n$ scaling pattern. Our experiments reveal a clear scaling relationship: when $K = 4k$, the optimal patch size is $p = 4$, while when $K = 16k$, the optimal patch size is $p = 16$, indicating that patch size and codebook size should be scaled together to achieve optimal performance. This scaling relationship reflects a balance between the computational pressure on the Patchify stage and the ViT blocks.

Our final model adopts latent dimension $d'$ consistent with the vector channel dimension (default 256), 2 ViT blocks, and $p = 16$ with a $K = 16k$ codebook. When scaling up the codebook, we simultaneously increase the patch size by the same factor of $4\times$. This scaling strategy ensures optimal performance across different model configurations while preserving computational efficiency.

## 4.3 SCALING STUDIES

Table 3 presents a comprehensive evaluation of scalability across different model configurations, examining the effects of codebook size, vector channels, and training epochs on reconstruction quality and codebook usage. The results demonstrate performance differences between VQGAN and FVQ across multiple scaling dimensions. All models are trained on ImageNet (Deng et al., 2009) and evaluated on both the ImageNet val set (50k) and COCO 2017 val set (5k) (LAION, 2022).

Table 3: **Results of scalability from codebook size, vector channel and training epoch.** $\text{rFID}_{\text{IN1K}}$ denotes the results evaluated on the ImageNet-1k validation set, while $\text{rFID}_{\text{COCO}}$ denotes the results evaluated on the COCO validation set. (-Ann.) denotes results without learning annealing.

| | Model | Training Epoch | Ratio | Codebook Size | Vector Channel | $\text{rFID}_{\text{IN1K}}$ | $\text{rFID}_{\text{COCO}}$ | Usage |
|---|---|---|---|---|---|---|---|---|
| 1 | VQGAN | 40 | 16 | 16k | 8 | 2.19 | 8.43 | 97% |
| 2 | VQGAN | 40 | 16 | 16k | 256 | 9.21 | – | 0.29% |
| 3 | FVQ$^{-\text{Ann.}}$ | 40 | 16 | 16k | 256 | 2.61 | – | 100% |
| 4 | FVQ | 40 | 16 | 16k | 4 | 1.71 | 7.64 | 100% |
| 5 | FVQ | 40 | 16 | 16k | 8 | 1.47 | 7.15 | 100% |
| 6 | FVQ | 40 | 16 | 16k | 256 | 1.30 | 6.84 | 100% |
| 7 | FVQ | 120 | 16 | 16k | 256 | 1.17 | 6.62 | 100% |
| 8 | FVQ | 40 | 16 | 65k | 256 | 1.14 | 6.39 | 100% |
| 9 | FVQ | 40 | 16 | 65k | 512 | 1.09 | 6.38 | 100% |
| 10 | FVQ | 120 | 16 | 65k | 512 | 1.00 | 6.16 | 100% |
| 11 | FVQ | 40 | 16 | 262k | 256 | 0.95 | 6.04 | 100% |
| 12 | FVQ | 120 | 16 | 262k | 256 | 0.88 | 5.83 | 100% |
| 13 | VQGAN | 40 | 8 | 16k | 8 | 0.98 | 5.03 | 97% |
| 14 | FVQ | 40 | 8 | 16k | 256 | 0.39 | 3.84 | 100% |

**Model Performance and Stability.** FVQ consistently outperforms VQGAN in both reconstruction quality and codebook usage. As shown in Figure 3, FVQ achieves 100% codebook usage early in training and maintains it throughout. Learning annealing further improves alignment in VQNs (from 2.61 to 1.30), as noted in **Observation 2**, which is also confirmed by our experiments. FVQ sustains full codebook usage across all tested configurations, whereas VQGAN experiences severe codebook collapse in high-capacity settings, with usage dropping to only 0.29% when employing 256 vector channels, validating **Observation 3**. This stability advantage is crucial for practical applications, as it ensures reliable performance scaling without catastrophic failure modes.

**Codebook Size Scaling.** Our experiments reveal clear scaling benefits with increased codebook capacity. For FVQ with 40 training epochs and 256 vector channels, expanding the codebook from 16k to 262k entries yields substantial improvements in reconstruction quality, with $\text{rFID}_{\text{IN1K}}$ decreasing from 1.30 to 0.95 and $\text{rFID}_{\text{COCO}}$ from 6.84 to 6.04. This trend indicates that larger codebooks provide enhanced representational capacity without suffering from underutilization issues.

**Vector Channel Effects.** Increasing vector channels from 4 to 256 leads to substantial improvements in reconstruction quality across both datasets. On ImageNet, $rFID_{IN1K}$ decreases from 1.71 to 1.30, while on COCO, $rFID_{COCO}$ improves from 7.64 to 6.84. This trend demonstrates that higher-dimensional vector representations enable more expressive feature encoding, allowing the model to capture finer visual details and achieve better reconstruction fidelity. Importantly, these findings provide a crucial foundation for integrating a semantic tokenizer (*e.g.*, CLIP (Radford et al., 2021) with 512-dimensional embeddings) with vector quantization.

**Training Duration Impact.** Extending training from 40 to 120 epochs yields consistent but diminishing returns on ImageNet, while on COCO it results in steady improvements, with $rFID_{COCO}$ increasing by approximately 0.2 at each evaluation interval.

**Compression Trade-offs.** Higher compression ratios (e.g., $16\times$) make the reconstruction task more challenging, reducing reconstruction quality, while lower compression ratios (e.g., $8\times$) facilitate learning and improve reconstruction fidelity. Both models benefit from reduced compression, but FVQ maintains its stability advantage across compression levels, achieving $rFID_{IN1K}$ of 0.39 and $rFID_{COCO}$ of 3.84 at $8\times$ compression with full codebook usage.

These findings establish FVQ as a more scalable and reliable architecture for vector quantization tasks, demonstrating predictable performance improvements with increased computational resources while maintaining full codebook usage across diverse configurations.

## 4.4 GENERALIZATION STUDIES

Recently, several multi-code representation methods have been proposed to improve VQ, where multiple tokens are used to represent a single spatial location (*e.g.*, RQ-VAE (Lee et al., 2022) and VAR (Tian et al., 2024)). To demonstrate the generalization ability of our approach, we further conduct experiments on these two methods.

Table 4: **Multi-Code Representation VQ.** $680_{10step}$ denotes a code map constructed through 10 sequential quantization steps with step sizes 1, 2, 3, 4, 5, 6, 8, 10, 13, 16, where 680 is the sum of squares of the sequence.

| Method | Training Epoch | Token Map | Codebook Size | Vector Channel | rFID$\downarrow$ |
|---|---|---|---|---|---|
| RQ-VAE (Lee et al., 2022) | 10 | $8 \times 8 \times 4$ | 16,384 | 256 | 4.73 |
| RQ-VAE (Lee et al., 2022) | 50 | $8 \times 8 \times 4$ | 16,384 | 256 | 3.20 |
| FVQ (RQ-VAE) | 10 | $8 \times 8 \times 4$ | 16,384 | 256 | 2.98 |
| VAR (Tian et al., 2024) | 40 | $680_{10step}$ | 4,096 | 32 | 1.00 |
| FVQ (VAR) | 40 | $680_{10step}$ | 16,384 | 256 | 0.80 |

We follow the original training protocols as closely as possible. The results are reported in Table 4, where our method consistently improves performance. Notably, with only 10 epochs of training, our approach surpasses RQ-VAE trained for 50 epochs. This highlights the strong generalization ability and training efficiency of our method.

## 5 MORE DISCUSSION

In this section, we clarify *why a shared MLP projector collapses under learning-rate annealing, whereas VQBridge remains stable*. We focus on the structural limitations of the MLP under straight-through estimation and contrast them with the dense, cross-code interactions introduced by VQBridge.

**MLP + STE leads to low-rank, usage-sparse updates.** In the baseline, all code vectors are transformed by a shared MLP $f_\theta$:

$$\hat{\mathbf{c}}_k = f_\theta(\mathbf{c}_k), \quad k = 1, \dots, K. \tag{12}$$

Although gradients from active codes update the shared parameters, only codes in the active set $A_t$ contribute to the STE loss at iteration $t$. The parameter gradient is

$$\frac{\partial L}{\partial \theta} = \sum_{k \in A_t} J(\mathbf{c}_k)^\top \mathbf{g}_k, \tag{13}$$

where $J(\mathbf{c}_k) = \partial f_\theta(\mathbf{c}_k)/\partial \theta$ is the Jacobian. Since $|A_t| \ll K$ for large codebooks, this update lies in a low-dimensional subspace defined by $\{J(\mathbf{c}_k)\}_{k \in A_t}$. Inactive codes are only indirectly updated by

$$\Delta \hat{\mathbf{c}}_k \approx J(\mathbf{c}_k)\, \Delta\theta_t, \tag{14}$$

which are typically weak (due to small cross-Jacobian interactions) and poorly aligned with the data regions those codes should represent.

**Why learning-rate annealing exacerbates collapse.** The usage of a code is determined by nearest-neighbor assignments. Usage changes only when a code crosses a Voronoi boundary. Before annealing, relatively large and noisy updates sometimes make such boundary crossings possible. However, once the learning rate decays,

$$\Delta\theta_t = -\eta_t \sum_{i \in A_t} J(\mathbf{c}_i)^\top \mathbf{g}_i, \qquad \eta_t \downarrow 0, \tag{15}$$

the induced changes on inactive codes,

$$\Delta \hat{\mathbf{c}}_k \approx J(\mathbf{c}_k)\, \Delta\theta_t, \tag{16}$$

become extremely small and resemble noise. These updates are insufficient to shift a code across any Voronoi boundary, so its nearest-neighbor assignments remain unchanged and its usage stays zero. The optimization therefore converges to a degenerate state where only a subset of codes is active, leading to codebook collapse.

**Why VQBridge remains stable.** VQBridge avoids this failure mode by introducing explicit cross-code interaction. The codebook $C \in \mathbb{R}^{K \times d}$ is first patchified, processed by several ViT blocks with global self-attention, and then unpatchified to produce $\hat{C} = f_\theta(C)$. Each output code aggregates information from multiple input codes, and gradients propagate through dense attention weights:

$$\frac{\partial L}{\partial \theta} \propto \sum_{k,j} \alpha_{kj}\, J_{kj}^\top \mathbf{g}_j, \tag{17}$$

where $\alpha_{kj}$ denotes attention weights. This mechanism directly transfers strong gradients from frequently used codes to many other codes within and across patches. The resulting updates are dense, globally coupled, and aligned with data geometry. Even under learning-rate annealing, inactive codes receive meaningful gradients, allowing them to re-enter active usage and preventing collapse.

**Summary.** The shared MLP projector collapses under learning-rate annealing because its pointwise architecture and STE produce inherently low-rank, usage-sparse updates that cannot revive inactive codes once the learning rate becomes small. VQBridge introduces structured cross-code interaction via self-attention, enabling dense and data-aligned gradient flow across the entire codebook. This design keeps all codes trainable and maintains near-complete codebook usage throughout training.

## 6    CONCLUSION

We have systematically analyzed the challenges inherent in discrete tokenizers and identified the root causes of unstable training and codebook collapse. Based on these insights, we proposed FVQ with the VQBridge projector, enabling robust, scalable, and high-quality VQ optimization. Our experiments show that high codebook usage and state-of-the-art reconstruction are achievable across diverse settings, and that strong tokenizers are essential for effective autoregressive image generation. Beyond improving current models, our work provides a foundation for extending VQ training to larger-scale and potentially unlabeled datasets. We believe these findings will inform future research on autoregressive visual generation and contribute to the development of more capable multimodal learning systems.

# 7 REPRODUCIBILITY STATEMENT

Section 3.2 describes the detailed design and implementation of FVQ. Appendix G provides comprehensive information on the experimental parameter settings, which can help readers reproduce the results reported in this paper. In addition, all datasets, preprocessing steps, and training protocols are clearly documented in the main text and supplementary materials to facilitate reproducibility. To further ensure reproducibility, we will release all source code, models, and configuration files immediately after the review process. These resources will allow other researchers to replicate our experiments and validate our findings.

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

# A    RELATED WORK

**Image Quantization.**    To convert image features into discrete tokens, VQ-VAE (Van Den Oord et al., 2017) maps encoded features to the nearest vectors in a learned codebook. VQ-VAE2 (Razavi et al., 2019) improves quantization with a hierarchical encoding strategy, and VQGAN (Esser et al., 2021) enhances perceptual quality via adversarial and perceptual losses. However, scaling the codebook size and vector channel often leads to codebook collapse—low usage and poor reconstruction. To mitigate this, ViT-VQGAN (Yu et al., 2022) reduces the vector channel and adds $l_2$ normalization to improve codebook usage. FSQ (Mentzer et al., 2023) and LFQ (Yu et al., 2024a) adopt similar low-channel designs to maintain usage with larger codebook. Reg-VQ (Zhang et al., 2023) introduces prior regularization, while CVQ (Zheng & Vedaldi, 2023) resets underused codebook entries. Yet, these methods still struggle when both size and channel are scaled, limiting reconstruction quality. Recently, (Huh et al., 2023) specifically analyzes the impact of the Straight-Through Estimator (Bengio et al., 2013). IBQ (Shi et al., 2025) achieves high usage with large codebook and high channel via index backpropagation. VQGAN-LC (Zhu et al., 2024a) initializes codebook with pretrained features and employs a linear projector to jointly transform embedding. SimVQ (Zhu et al., 2024b) further highlights the importance of joint transformation and provides theoretical support for linear mappings. We argue that scaling both the codebook size and vector channel requires strong projector to fully optimize the codebook and ensure effective usage.

**Image Generation.**    PixelCNN (Van den Oord et al., 2016) and iGPT (Chen et al., 2020) generates RGB pixels in a scanning order. VQGAN (Esser et al., 2021) improves (Van den Oord et al., 2016; Chen et al., 2020) by using an autoregressive approach in the latent space of VQVAE (Van Den Oord et al., 2017). VQVAE2 (Razavi et al., 2019) and RQ-Transformer (Lee et al., 2022) adopt the same approach but with additional scales. MaskGIT (Chang et al., 2022) employs masked prediction similar to BERT (Devlin et al., 2018). LlamaGen (Sun et al., 2024) demonstrates that autoregressive generation is possible even without visual bias, while VAR (Tian et al., 2024) explores visual autoregressive generation using multi-scale methods. RandAR (Pang et al., 2024) trains transformer in random orders to improve performance. Additionally, diffusion models (Ho et al., 2020; Song et al., 2020; Lipman et al., 2022; Rombach et al., 2022a;b; Peebles & Xie, 2023) generate images by denoising from noise. MAR (Li et al., 2024b) uses diffusion loss to eliminate the vector quantization process.

# B    LIMITATIONS AND FUTURE WORK

We primarily focus on the VQVAE/VQGAN-based learning paradigm to validate the effectiveness of our method. The solutions to codebook collapse and unstable training are general and can be further applied to models like UniTok (Ma et al., 2025) for additional improvements. Additionally, we only use LlamaGen as the generator; employing other strategies, such as random order learning and parallel decoding, can further enhance the generative capabilities of AR models.

**Pretraining Tokenizer.**    We plan to train more advanced discrete tokenizers on larger-scale datasets, such as OpenImages (Kuznetsova et al., 2020) and LAION-COCO (LAION, 2022), to further enhance the reconstruction capabilities of tokenizers and create better conditions for autoregressive generation.

**Unified Tokenizer.**    Our results demonstrate that the vector channel can be scaled to larger dimensions. This facilitates alignment with visual representation tokenizers like CLIP (Radford et al., 2021) (dim=512), enabling a unified tokenizer to accomplish both generative and understanding tasks.

**Generator Capacity.** Larger codebooks increase the parameter count of the generator's final layer, which can make optimization more challenging. A more balanced approach is to scale up the generator accordingly. Due to limited computational resources, we have not explored substantially larger codebooks and generators, which remains future work.

# C    ADDITIONAL EXPERIMENTS IN OBSERVATION

In Figure 7, VQ loss tends to stabilize at a lower value as the codebook size increases. Early in training, the loss drops quickly because only a few code vectors are active. This allows the encoder

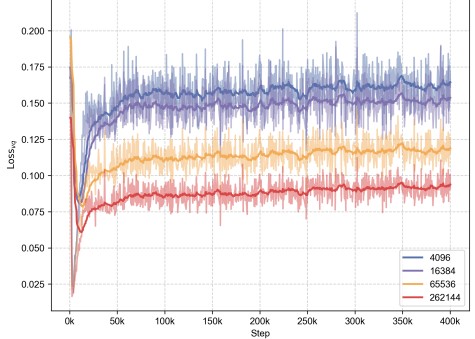

Figure 7: **Effect of codebook size on training loss**

Table 5: **Additional observational results.**

| Method | Codebook Size | usage | rFID |
|---|---|---|---|
| linear | 16k | 100% | 2.10 |
| linear+Annealing | 16k | 53% | 1.52 |
| VQBridge | 16k | 100% | 2.61 |
| VQBridge+Annealing | 16k | 100% | 1.30 |
| MLP-5 | 262k | 20% | 2.06 |
| MLP-5+Annealing | 262k | 22% | 1.75 |
| VQBridge | 262k | 100% | 1.40 |
| VQBridge+Annealing | 262k | 100% | 0.95 |

to fit the codebook entries easily. Additional results are reported in Table 5, which demonstrate the necessity of using VQBridge together with annealing.

## D DERIVATION OF THE ONE-STEP-BEHIND FORMULA

Consider the $t$-th iteration of VQ training with the following components: encoder output $\mathbf{z}_e^{(t)} \in \mathbb{R}^d$, quantized output $\mathbf{z}_q^{(t)}$.

The Vector Quantization commitment loss is defined as:

$$\mathcal{L}_{\text{vq}}\left(\mathbf{z}_e^{(t)}, \mathbf{z}_q^{(t)}\right) = \frac{1}{2}\left\|\mathbf{z}_e^{(t)} - \mathbf{z}_q^{(t)}\right\|_2^2. \tag{18}$$

The gradient of the commitment loss with respect to the vector code $\mathbf{z}_q$ is:

$$\frac{\partial \mathcal{L}_{\text{vq}}\left(\mathbf{z}_e^{(t)}, \mathbf{z}_q^{(t)}\right)}{\partial \mathbf{z}_q^{(t)}} = \frac{1}{2}\cdot 2\left(\mathbf{z}_q^{(t)} - \mathbf{z}_e^{(t)}\right). \tag{19}$$

The update rule for $\mathbf{z}_q^{(t+1)}$ when optimizing with SGD is:

$$\mathbf{z}_q^{(t+1)} = \mathbf{z}_q^{(t)} - \eta \frac{\partial \mathcal{L}_{\text{vq}}\left(\mathbf{z}_e^{(t)}, \mathbf{z}_q^{(t)}\right)}{\partial \mathbf{z}_q^{(t)}}. \tag{20}$$

Substituting the gradient:

$$\mathbf{z}_q^{(t+1)} = \mathbf{z}_q^{(t)} - \eta \cdot \left(\mathbf{z}_q^{(t)} - \mathbf{z}_e^{(t)}\right). \tag{21}$$

Expanding and rearranging:

$$\mathbf{z}_q^{(t+1)} = \mathbf{z}_q^{(t)} - \eta \mathbf{z}_q^{(t)} + \eta \mathbf{z}_e^{(t)} = (1 - \eta) \cdot \mathbf{z}_q^{(t)} + \eta \cdot \mathbf{z}_e^{(t)}. \tag{22}$$

From the above, we arrive at Equation 6. The correct update is given by Equation 7, and we derive an executable formula following (Huh et al., 2023). First,

$$\mathbf{z}_e^{(t+1)} = \mathbf{z}_e^{(t)} - \eta \left.\frac{\partial \mathcal{L}_{\text{task}}}{\partial \mathbf{z}_e}\right|_t. \tag{23}$$

Substituting Equation 23 into Equation 7:

$$\mathbf{z}_q^{(t+1)} = (1 - \eta) \cdot \mathbf{z}_q^{(t)} + \eta \cdot \mathbf{z}_e^{(t+1)} \tag{24}$$

$$= (1 - \eta) \cdot \mathbf{z}_q^{(t)} + \eta \left(\mathbf{z}_e^{(t)} - \eta \left.\frac{\partial \mathcal{L}_{\text{task}}}{\partial \mathbf{z}_e}\right|_t\right) \tag{25}$$

$$= \underbrace{(1 - \eta) \cdot \mathbf{z}_q^{(t)} + \eta \cdot \mathbf{z}_e^{(t)}}_{\text{commitment loss}} - \eta^2 \left.\frac{\partial \mathcal{L}_{\text{task}}}{\partial \mathbf{z}_e}\right|_t. \tag{26}$$

It can be observed that the additional term is multiplied by $\eta^2$. When annealing is applied, the contribution of this term is negligible, which underscores the importance of learning annealing.

# E  EFFECT OF ONE-STEP-BEHIND ON TRAINING STABILITY

In the $(t+1)$-th forward pass, quantization uses the *updated* codebook:

$$x^{(t+1)} = \arg\min_k \|\mathbf{z}_e^{(t+1)} - \mathbf{c}_k^{(t)}\|_2^2, \tag{27}$$

$$\mathbf{z}_q^{(t+1)} = \mathbf{c}_{x^{(t+1)}}^{(t)}. \tag{28}$$

Comparing with the quantization point used for backpropagation in step $t$: $\mathbf{z}_q^{(t)} = \mathbf{c}_{x^{(t)}}^{(t-1)}$, we observe a temporal misalignment: **Step $t$** encoder updates using signals around $\mathcal{C}^{(t-1)}$; **Step $t+1$ forward** quantization centers have changed to $\mathcal{C}^{(t)}$. This inconsistency constitutes **one-step-behind problem**.

**Mathematical Analysis.** The quantization center shift can be expressed as:

$$\underbrace{\mathbf{z}_q^{(t+1)} - \mathbf{z}_q^{(t)}}_{\text{Quantization center shift}} = \mathbf{c}_{x^{(t+1)}}^{(t)} - \mathbf{c}_{x^{(t)}}^{(t-1)}. \tag{29}$$

This decomposes into two components:

$$= \underbrace{\left(\mathbf{c}_{x^{(t+1)}}^{(t)} - \mathbf{c}_{x^{(t+1)}}^{(t-1)}\right)}_{\text{Codebook update effect}} + \underbrace{\left(\mathbf{c}_{x^{(t+1)}}^{(t-1)} - \mathbf{c}_{x^{(t)}}^{(t-1)}\right)}_{\text{Cluster switching}}. \tag{30}$$

By the triangle inequality,

$$\|\mathbf{z}_q^{(t+1)} - \mathbf{z}_q^{(t)}\| \leq \|\mathbf{c}_{x^{(t+1)}}^{(t)} - \mathbf{c}_{x^{(t+1)}}^{(t-1)}\| + \|\mathbf{c}_{x^{(t+1)}}^{(t-1)} - \mathbf{c}_{x^{(t)}}^{(t-1)}\|. \tag{31}$$

**Bounded Change Analysis.** Assume bounded changes:

$$\|\mathbf{c}_i^{(t)} - \mathbf{c}_i^{(t-1)}\| \leq \delta_c \quad \forall i, \quad \|\mathbf{z}_e^{(t+1)} - \mathbf{z}_e^{(t)}\| \leq \delta_z. \tag{32}$$

Define the assignment margin at time $t$ as the difference between the distances to the nearest and second-nearest centers (evaluated w.r.t. $\{\mathbf{c}_i^{(t-1)}\}$):

$$m^{(t)} := \|\mathbf{z}_e^{(t)} - \mathbf{c}_{2\text{nd}}^{(t-1)}\| - \|\mathbf{z}_e^{(t)} - \mathbf{c}_{1\text{st}}^{(t-1)}\|. \tag{33}$$

Using the triangle inequality one obtains that if $m^{(t)} > 2\delta_z$ then the nearest-center assignment is preserved (i.e. $x^{(t+1)} = x^{(t)}$); conversely, when $m^{(t)} \leq 2\delta_z$ cluster switching may occur. In the no-switching case

$$\|\mathbf{z}_q^{(t+1)} - \mathbf{z}_q^{(t)}\| \leq \delta_c, \tag{34}$$

while in the worst case (arbitrary switching) we may only guarantee

$$\|\mathbf{z}_q^{(t+1)} - \mathbf{z}_q^{(t)}\| \leq \delta_c + D_{\max}, \tag{35}$$

where $D_{\max} := \max_{i,j} \|\mathbf{c}_i^{(t-1)} - \mathbf{c}_j^{(t-1)}\|$ is a (time-local) upper bound on inter-center distances.

Finally, if the task loss $\mathcal{L}$ is $L$-Lipschitz w.r.t. $\mathbf{z}_q$, then

$$\left|\mathcal{L}(\mathbf{z}_q^{(t+1)}) - \mathcal{L}(\mathbf{z}_q^{(t)})\right| \leq L\|\mathbf{z}_q^{(t+1)} - \mathbf{z}_q^{(t)}\|. \tag{36}$$

Combining the above yields the following interpretation: when encoder outputs lie away from cluster boundaries ($m^{(t)} > 2\delta_z$) and codebook updates are small ($\delta_c$ small), loss fluctuations induced by one-step-behind are controlled by $L\delta_c$; however, near boundaries cluster switching can produce center jumps of order inter-cluster distances, which the Lipschitz property amplifies into large loss fluctuations and thus destabilizes training.

**Implication for the Decoder.** Since the quantized vector $\mathbf{z}_q$ is directly fed into the decoder, fluctuations in $\mathbf{z}_q$ propagate to the reconstruction stage. In the stable case (no switching, small $\delta_c$), the decoder receives smoothly varying inputs and can learn a consistent mapping. In contrast, when cluster switching occurs, $\mathbf{z}_q$ may undergo discontinuous jumps of order $D_{\max}$, forcing the decoder to fit inconsistent latent representations across consecutive steps. This misalignment not only destabilizes optimization but also degrades reconstruction quality, as the decoder struggles to reconcile abrupt changes in its input space.

## F  VECTOR QUANTIZATION FOR IMAGE RECONSTRUCTION

A discrete tokenizer contains Encoder, Codebook and Decoder. Given an image $\mathbf{I} \in \mathbb{R}^{H \times W \times 3}$, the Encoder compresses it into an image feature map $\mathbf{Z} \in \mathbb{R}^{h \times w \times D}$, where $H \times W$ and $h \times w$ denote the spatial resolution of the image and feature map, $D$ denotes the channel number of feature map. Formally,

$$\mathbf{Z} = \text{Encoder}(\mathbf{I}). \tag{37}$$

The Codebook contains $K$ vectors $\mathcal{C} = \{\mathbf{c}_1, \mathbf{c}_2, ..., \mathbf{c}_K\}$, where each vector $\mathbf{c}_i$ is a $D$-dimensional vector. By searching for the most similar vector in the codebook, $\mathbf{Z}_e$ is mapped to a feature $\mathbf{Z}_q$, and the indices form the token map $\mathbf{T}$. We denote the search process as $\text{Quant}(\cdot, \cdot)$. We have

$$\mathbf{c}_x = \text{Quant}(\mathbf{c}^{i,j}, \mathcal{C}), \tag{38}$$

where $x = \arg\min_{k \in \{1,2,...,K\}} \|\mathbf{z}_e^{i,j} - \mathbf{c}_k\|_2$, and

$$\mathbf{Z}_q = \{\mathbf{c}_t^{i,j}\}, \mathbf{X} = \{x^{i,j}\}, \tag{39}$$

where $i \in \{1, ..., h\}, j \in \{1, ..., w\}$. The Decoder reconstructs $\mathbf{Z}_q$ into $\hat{\mathbf{I}}$. Formally,

$$\hat{\mathbf{I}} = \text{Decoder}(\hat{\mathbf{Z}}), \tag{40}$$

Typically, a compound loss $\mathcal{L}$ is optimized to train the model. We have

$$\mathcal{L} = \mathcal{L}_{rec} + \mathcal{L}_{commit} + \lambda_{perc}\mathcal{L}_{perc} + \lambda_G\mathcal{L}_G, \tag{41}$$

where $\mathcal{L}_{rec} = \|\mathbf{I} - \hat{\mathbf{I}}\|_2$, $\mathcal{L}_{commit} = \|sg[\mathbf{Z}_e] - \mathbf{Z}_q\|_2 + \beta\|sg[\mathbf{Z}_q] - \mathbf{Z}_e\|_2$, $\mathcal{L}_{perc}$ is a perceptual loss like LPIPS (Zhang et al., 2018), $\mathcal{L}_G$ is a discriminative loss like PatchGAN (Isola et al., 2017), $\beta$, $\lambda_{perc}$, $\lambda_G$ are loss weights, and $sg[\cdot]$ is stop-gradient operation.

## G  FURTHER DETAILS ON EXPERIMENTAL CONFIGURATIONS

Table 6: **Experimental Configurations** on reconstruction and generation. Warmup(0.1) indicates that 10% of the training time is used for learning rate warmup.

| Config | Reconstruction | Generation |
| --- | --- | --- |
| Base Batch Size | 128 | 256 |
| Training Epochs | 40 or 120 | 300–350 |
| Optimizer | Adam | AdamW |
| Base Learning Rate | $1 \times 10^{-4}$ | $1 \times 10^{-4}$ |
| Learning Rate Schedule | warmup(0.1) | warmup(0) |
| | [baselr(0.3)–linear(0.7)] | [baselr(0.05)–linear(0.95)] |
| Warmup Learning Rate | 0.005 to 1 | − |
| Linear Learning Rate | 1 to 0.01 | 1 to 0.01 |
| Optimizer Parameters | $\beta_1 = 0.9, \beta_2 = 0.95$ | $\beta_1 = 0.9, \beta_2 = 0.95$ |
| | | $weight\_decay = 5 \times 10^{-2}$ |
| Loss Weight | commit loss $\beta = 0.25$ | |
| | perceptual loss $\lambda_{perc} = 1.0$ | |
| | discriminative loss $\lambda_G = 0.5$ | |

The training recipe Table 6 follows LlamaGen (Sun et al., 2024), where the learning rate is calculated as (batch size / base batch size) × base learning rate. The learning rate schedule specifies the proportions of different phases: warmup (0.1) indicates 10% of the training epochs, and [baselr(0.3)-linear(0.7)] represents the remaining 90%, with 30% maintaining a constant learning rate and 70% applying linear decay. All training is conducted on A100 GPUs, and evaluation follows the LlamaGen protocol. We use Fréchet Inception Distance (FID) as the primary metric, sampling 50,000 images and evaluating FID using code from ADM (Dhariwal & Nichol, 2021). Additionally, we report Inception Score (IS), Precision, and Recall.

# H FURTHER RESULTS COMPARISON

**Additional reconstruction results.** We present results for more metrics, including RSNR and SSIM, in Table 7. We report additional comparison results in Table 8, highlighting the superiority of FVQ. As shown in the Table 8, FVQ achieves the best reconstruction performance (rFID=0.88) among all discrete tokenizers, outperforming Open-MAGVIT2 and IBQ, even though they utilize larger architectures.

Table 7: **Additional Metrics and Comparisons.** [‡] trained for 120 epochs.

| Method | Codebook Size | rFID↓ | RSNR↑ | SSIM↑ | Usage |
|---|---|---|---|---|---|
| LlamaGen (Sun et al., 2024) | 16,384 | 2.19 | 20.79 | 0.675 | 97.0% |
| FVQ | 16,384 | 1.30 | 22.36 | 0.648 | 100% |
| FVQ[‡] | 16,384 | 1.17 | 22.45 | 0.653 | 100% |
| FVQ | 262,144 | 0.95 | 22.90 | 0.665 | 100% |
| FVQ[‡] | 262,144 | 0.88 | 22.95 | 0.672 | 100% |

Table 8: **Reconstruction performance of different tokenizers on 256 × 256 ImageNet 50k validation set.** [*] trained on OpenImages (Kuznetsova et al., 2020). [†] trained for 330 epochs. [‡] trained for 120 epochs. [◇] trained with 4 ResBlocks.

| Method | Tokens | Codebook Size | Vector Channel | rFID↓ | LPIPS↓ | Codebook Usage↑ |
|---|---|---|---|---|---|---|
| VAE-SD[*] (Rombach et al., 2022a) | 16 × 16 | | 16 | 0.87 | – | – |
| VAE-MAR (Li et al., 2024b) | 16 × 16 | – | 16 | 1.22 | – | – |
| VAVAE (Yao et al., 2025) | 16 × 16 | – | 16 | 0.28 | – | – |
| *VQ-VAE-based Methods* | | | | | | |
| VQGAN (Esser et al., 2021) | 16 × 16 | 1,024 | 256 | 7.94 | – | 44% |
| VQGAN (Esser et al., 2021) | 16 × 16 | 16,384 | 256 | 4.98 | 0.17 | 5.9% |
| VQGAN-SD[*] (Rombach et al., 2022a) | 16 × 16 | 16,384 | 8 | 5.15 | – | – |
| MaskGIT (Chang et al., 2022) | 16 × 16 | 1,024 | 256 | 2.28 | – | – |
| LlamaGen (Sun et al., 2024) | 16 × 16 | 16,384 | 8 | 2.19 | 0.14 | 97% |
| LlamaGen (Sun et al., 2024) | 16 × 16 | 16,384 | 256 | 9.21 | – | 0.29% |
| VQGAN-LC (Zhu et al., 2024a) | 16 × 16 | 16,384 | 8 | 3.01 | 0.13 | 99% |
| VQGAN-LC (Zhu et al., 2024a) | 16 × 16 | 100,000 | 8 | 2.62 | 0.12 | 99% |
| Open-MAGVIT2[†◇] (Luo et al., 2024) | 16 × 16 | 16,384 | 0 | 1.58 | 0.23 | 100% |
| Open-MAGVIT2[†◇] (Luo et al., 2024) | 16 × 16 | 262,144 | 0 | 1.17 | 0.20 | 100% |
| IBQ[†] (Shi et al., 2025) | 16× 16 | 16,384 | 256 | 1.55 | 0.23 | 97% |
| IBQ[†◇] (Shi et al., 2025) | 16 × 16 | 16,384 | 256 | 1.37 | 0.22 | 96% |
| IBQ[†◇] (Shi et al., 2025) | 16 × 16 | 262,144 | 256 | 1.00 | 0.20 | 84% |
| FVQ | 16 × 16 | 16,384 | 256 | 1.30 | 0.13 | 100% |
| FVQ[‡] | 16 × 16 | 16,384 | 256 | 1.17 | 0.13 | 100% |
| FVQ | 16 × 16 | 262,144 | 256 | 0.95 | 0.12 | 100% |
| FVQ[‡] | 16 × 16 | 262,144 | 256 | 0.88 | 0.12 | 100% |
| *1-D Token Sequence* | | | | | | |
| Titok-L (Yu et al., 2024b) | 32 | 4,096 | 16 | 2.21 | – | – |
| Titok-B (Yu et al., 2024b) | 64 | 4,096 | 16 | 1.70 | – | – |
| Titok-S (Yu et al., 2024b) | 128 | 4,096 | 16 | 1.71 | – | – |
| *Multi-Code Representation* | | | | | | |
| RQ-VAE$_{10ep}$ (Lee et al., 2022) | 8 × 8 × 4 | 16,384 | 256 | 4.73 | – | – |
| RQ-VAE$_{50ep}$ (Lee et al., 2022) | 8 × 8 × 4 | 16,384 | 256 | 3.20 | – | – |
| MoVQ (Zheng et al., 2022) | 16 × 16 × 4 | 1,024 | 64 | 1.12 | – | – |
| VAR (Tian et al., 2024) | $680_{10step}$ | 4,096 | 32 | 1.00 | – | – |
| FVQ (RQ-VAE$_{10ep}$) | 8 × 8 × 4 | 16,384 | 256 | 2.98 | – | – |
| FVQ (VAR) | $680_{10step}$ | 16,384 | 256 | 0.80 | – | – |

**Additional ablation experiments.** We perform addtional ablation experiments to study the effects of warmup and learning annealing. Our results indicate that warmup considerably influences performance, in line with the observations reported in (Huh et al., 2023). In contrast, the choice of learning annealing schedule—linear or cosine—has a negligible impact on the outcome. The detailed results are summarized in Table 9. Table 10 reports the computational cost of different projectors, highlighting the efficiency of VQBridge.

Table 9: **Additional Ablation Experiments.**

| Method | Codebook Size | rFID |
|---|---|---|
| baseline w/ linear | 16,384 | 1.30 |
| w/o warmup | 16,384 | 1.70 |
| w/ cosine | 16,384 | 1.32 |

Table 10: **Estimated Model Computational Cost (in FLOPs)**

| Codebook Size | Linear | MLP-5 | VQBridge |
|---|---|---|---|
| 16,384 | 1.07G | 5.37G | 4.30G |
| 262,144 | 17.18G | 85.90G | 55.89G |

**Additional generative results.** We report more additional generative results in Table 11. FVQ-XL (775M, 2.07) surpasses VAR-d24 (1.0B, 2.09), RandAR-XXL (1.4B, 2.15) and L-DiT (3.0B, 2.10; 7.0B, 2.28), further demonstrating the importance of tokenizers and the superiority of AR methods. FVQ can also further enhance the capabilities of other AR models. Due to time and computational constraints, we plan to extend this work in the future.

Table 11: **Generative model comparison on class-conditional ImageNet $256 \times 256$.** The metrics include Fréchet Inception Distance (FID), Inception Score (IS), Precision (Pre), and Recall (Rec). ‡ trained with a learning rate schedule.

| Type | Model | #Para | FID↓ | IS↑ | Pre↑ | Rec↑ |
|---|---|---|---|---|---|---|
| GAN | BigGAN (Brock et al., 2018) | 112M | 6.95 | 224.5 | 0.89 | 0.38 |
| GAN | GigaGGAN (Kang et al., 2023) | 569M | 3.45 | 225.5 | 0.84 | 0.61 |
| GAN | StyleGan-XL (Sauer et al., 2022) | 166M | 2.30 | 265.1 | 0.78 | 0.53 |
| Diff. | ADM (Dhariwal & Nichol, 2021) | 554M | 10.94 | 101.0 | 0.69 | 0.63 |
| Diff. | CDM (Ho et al., 2022) | – | 4.88 | 158.7 | – | – |
| Diff. | LDM-4-G (Rombach et al., 2022a) | 400M | 3.60 | 247.7 | – | – |
| Diff. | DiT-L/2 (Peebles & Xie, 2023) | 458M | 5.02 | 167.2 | 0.75 | 0.57 |
| Diff. | DiT-XL/2 (Peebles & Xie, 2023) | 675M | 2.27 | 278.2 | 0.83 | 0.57 |
| Diff. | L-DiT-3B (Alpha-VLLM, 2024) | 3.0B | 2.10 | 304.4 | 0.82 | 0.60 |
| Diff. | L-DiT-7B (Alpha-VLLM, 2024) | 7.0B | 2.28 | 316.2 | 0.83 | 0.58 |
| Diff. | MaskDiT (Zheng et al., 2023) | 675M | 2.28 | 276.6 | 0.80 | 0.61 |
| Diff. | SiT (Ma et al., 2024) | 675M | 2.06 | 270.3 | 0.82 | 0.59 |
| Mask. | MaskGIT (Chang et al., 2022) | 227M | 6.18 | 182.1 | 0.80 | 0.51 |
| Mask. | RCG (Li et al., 2024a) | 502M | 3.49 | 215.5 | – | – |
| MAR | MAR-L (Li et al., 2024b) | 479M | 1.98 | 290.3 | – | – |
| MAR | MAR-H (Li et al., 2024b) | 943M | 1.55 | 303.7 | 0.81 | 0.62 |
| VAR | VAR-d16 (Tian et al., 2024) | 310M | 3.30 | 274.4 | 0.84 | 0.51 |
| VAR | VAR-d20 (Tian et al., 2024) | 600M | 2.57 | 302.6 | 0.83 | 0.56 |
| VAR | VAR-d24 (Tian et al., 2024) | 1.0B | 2.09 | 321.9 | 0.82 | 0.59 |
| VAR | VAR-d30 (Tian et al., 2024) | 2.0B | 1.92 | 323.1 | 0.82 | 0.59 |
| AR | VQGAN (Esser et al., 2021) | 227M | 18.65 | 80.4 | 0.78 | 0.26 |
| AR | ViTVQ (Yu et al., 2022) | 1.7B | 4.17 | 175.1 | – | – |
| AR | RQTran. (Lee et al., 2022) | 3.8B | 7.55 | 134.0 | – | – |
| AR | RandAR-L (Pang et al., 2024) | 343M | 2.55 | 288.9 | 0.81 | 0.58 |
| AR | RandAR-XL (Pang et al., 2024) | 775M | 2.25 | 317.8 | 0.80 | 0.60 |
| AR | RandAR-XXL (Pang et al., 2024) | 1.4B | 2.15 | 321.9 | 0.79 | 0.60 |
| AR | LlamaGen-L (Sun et al., 2024) | 343M | 3.81 | 248.3 | 0.83 | 0.52 |
| AR | LlamaGen-XL (Sun et al., 2024) | 775M | 3.39 | 227.1 | 0.81 | 0.54 |
| AR | LlamaGen-XXL (Sun et al., 2024) | 1.4B | 3.09 | 253.6 | 0.83 | 0.53 |
| AR | IBQ-B (Shi et al., 2025) | 342M | 2.88 | 254.4 | 0.84 | 0.51 |
| AR | IBQ-L (Shi et al., 2025) | 649M | 2.45 | 267.5 | 0.83 | 0.52 |
| AR | IBQ-XL (Shi et al., 2025) | 1.1B | 2.14 | 279.0 | 0.83 | 0.56 |
| AR | FVQ-L (LlamaGen-L) | 343M | 2.39 | 275.8 | 0.84 | 0.56 |
| AR | FVQ-XL (LlamaGen-XL) | 775M | 2.07 | 287.0 | 0.83 | 0.58 |

## I    FURTHER VISUALIZATION ANALYSIS

**Qualitative Reconstruction and Generation Samples.**    We present reconstruction samples in Figure 8 and generation samples in Figure 9-Figure 11.

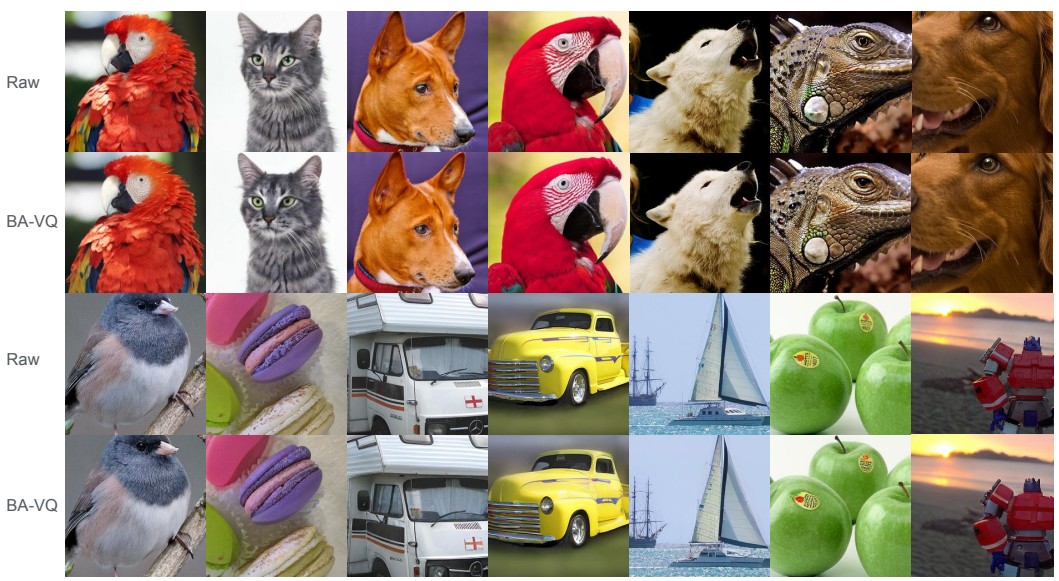

Figure 8: **FVQ-16,384 (rFID=1.17) reconstruction samples.**

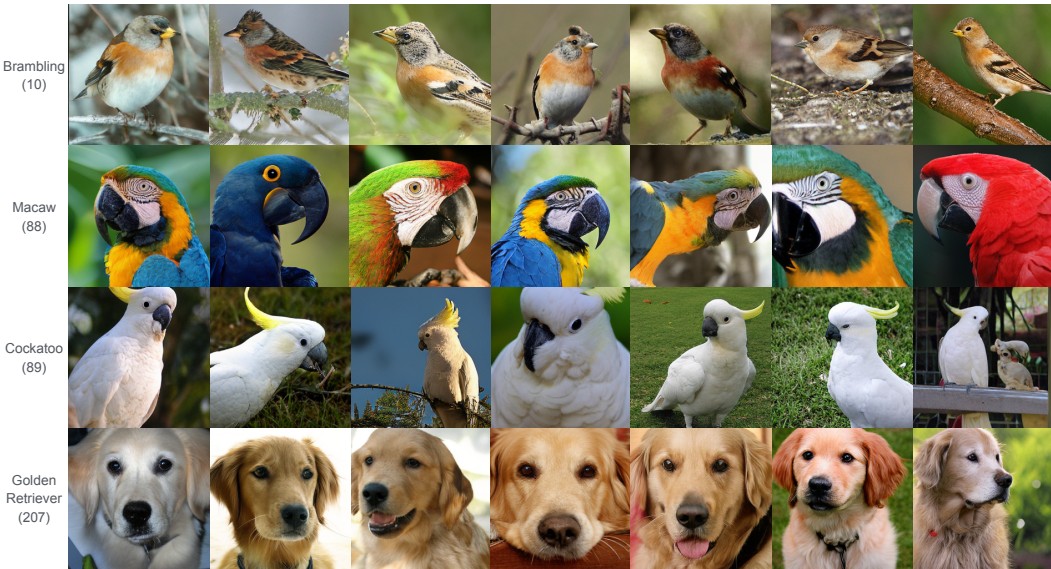

Figure 9: $256 \times 256$ **FVQ-XL (FID=2.07) samples.** Classifier-free guidance scale=1.65.

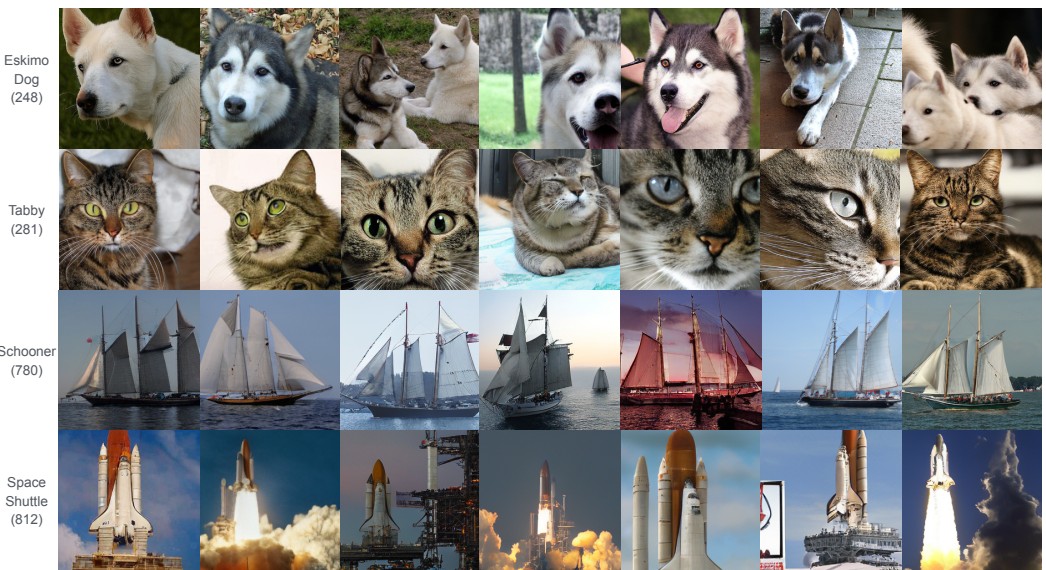

Figure 10: $256 \times 256$ **FVQ-XL (FID=2.07) samples.** Classifier-free guidance scale=1.65.

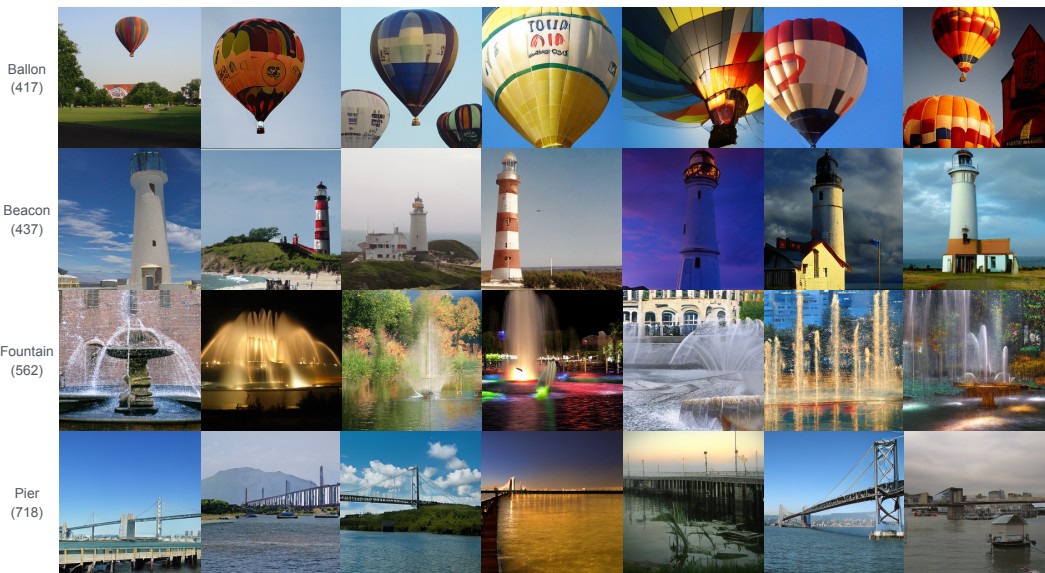

Figure 11: $256 \times 256$ **FVQ-XL (FID=2.07) samples.** Classifier-free guidance scale=1.65.

## J  TRAINING COST

We report additional details regarding training resources. The VQBridge module introduces only 3M extra parameters, accounting for approximately 4% of the total model size. This increase is minor and does not lead to additional computational burden. Furthermore, the overall training time remains effectively unchanged, aside from normal variations in server throughput. These results indicate that VQBridge provides performance gains with negligible training overhead.

Table 12: **Training Cost.** This table summarizes the parameter overhead and training time of different configurations. VQBridge introduces only a small number of additional parameters while keeping the overall training time effectively unchanged, indicating negligible computational overhead.

| Model | Module | Total Parameters | Extra Parameters | Training Time |
|---|---|---|---|---|
| VQGAN (Sun et al., 2024) | w/ nothing | 76.07M | 0M | ~27h |
| FVQ | w/ linear | 76.14M | 0.66M | ~27h |
| FVQ | w/ VQBridge | 79.23M | 3.15M | ~27h |

## K  LLM USAGE STATEMENT

In the preparation of this paper, a large language model (LLM) was only used as a general-purpose writing assistant to improve grammar, phrasing, and clarity. The LLM did not contribute to the research ideation, experimental design, analysis, or the generation of scientific content. All ideas, results, and conclusions presented in this paper are solely the work of the authors, who take full responsibility for the content.

