# OpenReview forum: "Scalable Training for Vector-Quantized Networks with 100% Codebook Utilization"
_ICLR.cc/2026/Conference — ICLR 2026 Poster_

### Official Review · Reviewer_pncL · 2025-10-24

**Soundness:** 2
**Presentation:** 2
**Contribution:** 2
**Rating:** 6
**Confidence:** 2

**Summary:**

This paper investigates the training challenges of vector quantization (VQ) in image generation, identifying straight-through estimation bias, one-step-behind updates, and sparse gradients as the root causes of instability and codebook collapse. To address these issues, the authors propose VQBridge, a projector with a “compress–process–recover” pipeline, and combine it with learning rate annealing to form FVQ. Experiments show that FVQ consistently achieves 100% codebook utilization, significantly improves reconstruction and generation quality, and enables autoregressive image models to outperform both existing autoregressive and diffusion models, while maintaining strong scalability and generalization.

**Strengths:**

1. Clearly identifies and analyzes the three core challenges of VQ training (STE bias, one-step-behind updates, sparse gradients) with both theoretical and empirical support.
2. Proposes a simple yet effective solution (VQBridge) that ensures 100% codebook utilization, addressing long-standing codebook collapse issues.
3. Strong experimental results: FVQ outperforms prior tokenizers in both reconstruction and generation quality, and even enables autoregressive models to surpass diffusion models.
4. Demonstrates robustness and scalability across codebook sizes (up to 262k), vector channels, and training epochs, with consistent improvements.
5. Shows generalization to multi-code VQ methods (RQ-VAE, VAR), suggesting broader applicability.

**Weaknesses:**

1. Novelty may be somewhat incremental, as the idea of using projectors (linear layers, joint optimization) has been explored; the main contribution is scaling with a stronger projector (ViT-based).
2. Experimental comparisons focus mainly on ImageNet and a few baselines; evaluation on more diverse datasets or real-world tasks would strengthen claims.
3. While FVQ achieves 100% codebook usage, the practical benefit of full utilization versus ~95–99% is not fully clarified.
4. The method’s additional training cost and complexity (e.g., ViT blocks in VQBridge) are not extensively analyzed.
5. Limited discussion of limitations or potential failure cases (e.g., extremely large-scale pretraining, cross-modal tokenizers).

**Questions:**

Please refer to the weakness section.

---

> ### Author Response · Authors · 2025-11-20
> **Rebuttal by Authors**
>
> Dear Reviewer pncL,
>
> We thank you for valuable comments and questions, which help us improve our work. Our responses are below.
>
> ---
>
> >**[W1] Clarifying Novelty**
>
> [A1] We appreciate the feedback regarding the novelty of our work. Our main contribution is a focused analysis of the core issues in vector quantization. Based on these observations, we introduce FVQ, a simple training framework that consistently achieves 100% codebook usage, improves reconstruction quality, scales with larger codebooks, and works across different VQ variants. We also show that FVQ significantly improves downstream generative modeling, where even a vanilla autoregressive model equipped with FVQ outperforms strong VAR and diffusion baselines. Although the required modifications are small, they directly address the underlying bottlenecks and therefore lead to substantial performance gains.
>
> >**[W2] Evaluations on datasets beyond ImageNet and comparisons with additional baselines.**
>
> [A2] We thank the reviewer for comprehensive comments regarding a more thorough evaluation. In addition to ImageNet, we also evaluate our method on the COCO dataset (see Table 3). These two datasets are the most commonly used benchmarks for image reconstruction and have been adopted by prior works such as Var [1], LlamaGen [2] and IBQ [3].
>
> In the main text, we compare our method against the baselines most relevant to the experimental setting. Additional baseline comparisons are provided in the appendix, with reconstruction performance reported in Table 8 and generation performance in Table 11. We have added the baselines VAVAE, SiT, and MaskDiT for comparison in the revision.
>
> >**[W3] While FVQ achieves 100% codebook usage, the practical benefit of full utilization versus ~95–99% is not fully clarified.**
>
> [A3] Thank you to the reviewer for the insightful and professional comments. Since all our configurations achieve 100% codebook usage, it is difficult to directly compare with the 95–99% range. To illustrate the benefit of full usage, we compare IBQ [3] and FVQ under the same configuration. The results clearly indicate that higher codebook usage has a noticeable positive impact on performance, such as rFID.
>
> | **Model** | **Codebook Size** | **Vector Dim** | **usage** |**rFID**|
> |------|----------|----------------|-------|----|
> | IBQ |16k | 256 | 97% | 1.55|
> | FVQ | 16k | 256 | 100% | 1.30 |
>
> >**[W4] The method’s additional training cost and complexity (e.g., ViT blocks in VQBridge) are not extensively analyzed.**
>
> [A4] Thank you for your suggestions to provide more details on computational cost.
> First, we provide the computational cost of the VQBridge module (measured in FLOPs) in Appendix H, Table 10.
>
> Additionally, we include more details regarding the training resources below. It is important to note that the extra 3M parameters introduced by VQBridge account for only about 4% of the total parameters, which is a very small increase. Moreover, the overall training time remains almost unchanged (considering normal fluctuations in server throughput).
>
> | **Model** | **Module** | **Total Parameters** | **Extra Parameters** | **Training Time** |
> |------|----------|----------------|-------|---------------|
> | VQGAN | w/ nothing | 76.07M | 0M | ~27h |
> | FVQ | w/ linear | 76.14M | 0.66M | ~27h |
> | FVQ | w/ VQBridge | 79.23M | 3.15M | ~27h |
>
> > **[W5] Clarifying limitations or potential failure cases**
>
> [A5] We discuss limitations and directions for future work in Appendix B, including pretraining tokenizers and  unified tokenizers. To demonstrate the potential of our approach, we conduct an additional experiment using more training data (IN1K + OpenImages). The results show consistent improvements in rFID, particularly when evaluated on the more diverse and realistic COCO dataset, indicating that our method benefits from larger and richer datasets.
>
> | **Model** | **Training Data** | **Traning Epoch** | **Codebook Size** | **Codebook Size** | rFID_IN1K | rFID_COCO|
> |------|----------|----------------|-------|---------------|----|----|
> | FVQ |IN1K | 120ep | 65k | 512 | 1.00 | 6.16 |
> | FVQ | IN1K+OpenI | 120ep | 65k | 512 | 0.95 | 5.93 |
>
> **References:**
>
> [1] Tian K, Jiang Y, Yuan Z, et al. Visual autoregressive modeling: Scalable image generation via next-scale prediction[J]. Advances in neural information processing systems, 2024, 37: 84839-84865.
>
> [2] Sun P, Jiang Y, Chen S, et al. Autoregressive model beats diffusion: Llama for scalable image generation[J]. arXiv preprint arXiv:2406.06525, 2024.
>
> [3] Shi F, Luo Z, Ge Y, et al. Scalable image tokenization with index backpropagation quantization[C]//Proceedings of the IEEE/CVF International Conference on Computer Vision. 2025: 16037-16046.
>
> ---
>
> **We thank the reviewer again for thorough and professional evaluation. We hope our responses address the reviewer’s concerns, and we welcome any further suggestions throughout the rebuttal period. Please feel free to share any additional questions or feedback!**

---

> > ### Comment · Reviewer_pncL · 2025-11-25
> >
> > Thank you for the clarifications. I am satisfied with the responses and will maintain my score.

---

> > > ### Author Response · Authors · 2025-11-26
> > > **Thank You for Your Follow-Up**
> > >
> > > Thank you for your kind follow-up. We appreciate your time and thoughtful feedback, and we’re glad our responses addressed your concerns.

---

### Official Review · Reviewer_93aA · 2025-10-29

**Soundness:** 3
**Presentation:** 3
**Contribution:** 3
**Rating:** 6
**Confidence:** 5

**Summary:**

This paper proposes a new framework for VQ-VAE called VQBridge, which enables robust and scalable codebook optimization. The proposed FVQ achieves 100% codebook usage for various codebook size. Extensive experiments demonstrate that FVQ outperforms prior methods and significantly boosts the generative performance of autoregressive models.

**Strengths:**

- This paper is well written and easy to follow.

- The proposed method achieves robust and scalable codebook optimization, enabling 100% codebook usage even with very large codebooks.

- The authors conduct various experiments to show the effectiveness of the proposed results, both quantitatively and qualitatively.

**Weaknesses:**

- While the empirical results are strong, the paper does not include an in-depth theoretical justification for why the proposed method achieves 100% codebook usage while VQGAN-LC cannot, especially considering that the only difference between VQGAN-LC and FVQ is the network architecture (MLP vs. ViT).

- The authors need to provide experiments using larger codebooks (e.g., 65k or 262k) for image generation to demonstrate the necessity of using a large codebook.

**Questions:**

- Why does learning rate annealing have such a significant impact on the MLP?

- Does using a large codebook pose challenges for generation?

---

> ### Author Response · Authors · 2025-11-20
> **Rebuttal by Authors**
>
> Dear Reviewer 93aA,
>
> We thank you for valuable comments and questions, which help us improve our work. Our responses are below.
>
> ---
>
> >**[W1, Q1] Why does learning rate annealing have such a significant impact on the MLP?**
>
> [A1] Thank you for your expert and valuable questions. We provide comprehensive answers in the overall author rebuttal. We have added this discussion in our revision (Sec 5 More Discussion).
>
>
> >**[W2, Q2] Using a larger codebook for image generation.**
>
> [A2] Thank you for this professional question. We evaluate FVQ models with different codebook sizes (4k, 16k, 65k) using LlamaGen-L as the generator. The results are summarized below:
>
> |  **Codebook Size**| 4k|16k| 65k |
> |------------|--------|---------|---|
> | **FID**    | 2.65 |2.39  | 2.30|
>
> As shown, increasing the codebook size consistently improves reconstruction and generation quality, consistent with observations from prior work such as IBQ [1]. However, a larger codebook also increases the parameter count of the generator’s final linear layer, which can make optimization more challenging. Simply adding a final linear layer may easily lead to model imbalance, and enlarging the overall model might be more effective. In general, larger codebooks require a larger-capacity generator to fully realize their benefits.
>
> Due to our limited training resources, we are currently unable to evaluate even larger codebook sizes with larger generators. We have included this limitation in the discussion in the revision and consider it part of our future work.
>
> **References:**
>
> [1] Shi F, Luo Z, Ge Y, et al. Scalable image tokenization with index backpropagation quantization[C]//Proceedings of the IEEE/CVF International Conference on Computer Vision. 2025: 16037-16046.
>
> ---
> **Thank you again for your insightful and professional comment, which helped us make our work more complete and solid! If you have any further questions, please feel free to let us know. We would be happy to provide additional clarifications.**

---

> ### Comment · Reviewer_93aA · 2025-11-25
>
> Thank you for the response. I am inclined to keep my original rating.

---

> > ### Author Response · Authors · 2025-11-26
> > **Thanks for your timely reply and the kind words!**
> >
> > Thank you for your quickly reply! We will remain active until the discussion period ends. Please feel free to get back to us if you have any new questions :-)!

---

### Official Review · Reviewer_wFm6 · 2025-10-30

**Soundness:** 3
**Presentation:** 4
**Contribution:** 3
**Rating:** 6
**Confidence:** 2

**Summary:**

This paper addresses the long-standing problems of training instability and codebook collapse in vector-quantized (VQ) networks, especially when scaling up the codebook size. The authors attribute these issues to three fundamental challenges: straight-through estimator (STE) bias, one-step-behind updates, and sparse codebook gradients. To solve this, the paper proposes a training framework called FVQ (FullVQ), featuring a novel, ViT-based projector named VQBridge. VQBridge uses a "compress-process-recover" pipeline to jointly optimize the codebook vectors. This method, combined with learning annealing, aims to enable stable and scalable training. Experimental results demonstrate that FVQ achieves 100% codebook usage across various configurations (up to 262k) and attains state-of-the-art image reconstruction performance (rFID 0.88). More importantly, FVQ significantly boosts the performance of downstream autoregressive (AR) image generation models, enabling LlamaGen-XL to outperform strong diffusion models like DiT-XL/2.

**Strengths:**

1. Significance: Addresses a core bottleneck in VQ training—codebook collapse—which is critical for scalable discrete representation learning.
 2. Effectiveness: The results are outstanding. Achieving 100% usage on a 262k codebook and an rFID of 0.88 is a major engineering and empirical achievement.
3. Downstream Impact: One of the paper's strongest points is demonstrating how a good tokenizer empowers AR models. By only swapping the tokenizer, LlamaGen-L's FID improved from 3.81 to 2.39, and LlamaGen-XL's from 3.39 to 2.07, surpassing DiT-XL/2 (2.27). This is a very compelling result.
4. Clarity: The paper is clearly written and well-organized. The problem, challenges, and solution are all well-articulated. 5. Thoroughness: The experimental evaluation is comprehensive, covering detailed architectural ablations, scaling analysis, and generalization tests.

**Weaknesses:**

1. Limited Novelty: The core idea of using a projector for joint codebook optimization is not entirely new. The paper itself cites prior work that used linear layers for this. The main novelty lies in the design of a much more powerful, DiT-inspired projector (VQBridge). While highly effective, this feels more like an excellent architectural improvement than a fundamental conceptual leap.
 2. Training Overhead: The VQBridge module introduces significant computational cost during training. Table 10 shows that for a 16k codebook, VQBridge has ~4x the FLOPS of a linear layer, and for 262k, it's ~3.2x. While the paper correctly states it adds no inference cost, this training overhead is a practical limitation that warrants more discussion.
3. Analysis of Annealing: The paper claims learning annealing is crucial and shows a large performance gain in Table 3. However, the analysis in Figure 3 is confusing: it shows a linear layer's usage drops significantly with annealing, while VQBridge's remains at 100%. This suggests VQBridge is simply robust to annealing, rather than it being a synergistic component. Why does annealing hurt the linear projector's usage so badly? This is not fully explained.

**Questions:**

1. (Re: Weakness 2) Could the authors elaborate on the practical training time (wall-clock) and VRAM overhead of VQBridge compared to baselines (e.g., VQGAN or VQGAN-LC w/ linear projector)? Table 10 gives FLOPS, but what is the practical impact for reproducibility and adoption?
2. (Re: Weakness 3) The interaction with learning annealing is interesting. In Figure 3, annealing drastically hurts the linear projector's usage but has no effect on VQBridge's. Yet, Table 3 shows annealing is vital for VQBridge's performance (rFID). Could the authors clarify this? Does VQBridge's powerful joint optimization prevent the collapse that annealing would otherwise cause in a weaker projector?
 3. The VQBridge design is inspired by DiT. Did the authors experiment with other powerful interaction mechanisms? It's surprising that a 5-layer MLP failed so badly (20% usage). Why do the authors believe the ViT block (Patchify -> Process -> Unpatchify) architecture is so much more effective than a deep MLP for this task?

---

> ### Author Response · Authors · 2025-11-20
> **Rebuttal by Authors**
>
> Dear Reviewer wFm6,
>
> We thank you for valuable comments and questions, which help us improve our work. Our responses are below.
>
> ---
>
> >**[W1] Clarifying Novelty**
>
> [A1] We appreciate the feedback regarding the novelty of our work. Our main contribution is a focused analysis of the core issues in vector quantization. Based on these observations, we introduce FVQ, a simple training framework that consistently achieves 100% codebook usage, improves reconstruction quality, scales with larger codebooks, and works across different VQ variants. We also show that FVQ significantly improves downstream generative modeling, where even a vanilla autoregressive model equipped with FVQ outperforms strong VAR and diffusion baselines. Although the required modifications are small, they directly address the underlying bottlenecks and therefore lead to substantial performance gains.
>
> >**[W2,Q1] Training Overhead**
>
> [A2] We appreciate your thorough comment. We report the training overhead using the codebook size of 16k as a representative example below. It is important to note that the extra 3M parameters introduced by VQBridge account for only about 4% of the total parameters, which is a very small increase. Moreover, the overall training time remains almost unchanged (considering normal fluctuations in server throughput). We have added this discussion in our revision (see Appenxid J Traning Cost).
>
> | **Model** | **Module** | **Total Parameters** | **Extra Parameters** | **Training Time** |
> |------|----------|----------------|-------|---------------|
> | VQGAN | w/ nothing | 76.07M | 0M | ~27h |
> | FVQ | w/ linear | 76.14M | 0.66M | ~27h |
> | FVQ | w/ VQBridge | 79.23M | 3.15M | ~27h |
>
> >**[W3,Q2] In Figure 3, annealing drastically hurts the linear projector's usage but has no effect on VQBridge's. Yet, Table 3 shows annealing is vital for VQBridge's performance (rFID). Could the authors clarify this? Does VQBridge's powerful joint optimization prevent the collapse that annealing would otherwise cause in a weaker projector?**
>
> [A3] Yes, VQBridge demonstrates stronger robustness, which prevents collapse under learning rate annealing. We provide additional experimental results in Appendix C, Table 5. The purpose of annealing is to mitigate the effects of Challenge 2 (the one-step-behind update). The strong VQBridge architecture successfully avoids collapse during annealing. For additional details, please refer to the overall author rebuttal.
>
> >**[Q3] The VQBridge design is inspired by DiT. Did the authors experiment with other powerful interaction mechanisms? It's surprising that a 5-layer MLP failed so badly (20% usage). Why do the authors believe the ViT block (Patchify -> Process -> Unpatchify) architecture is so much more effective than a deep MLP for this task?**
>
> [A4] We appreciate your feedback and respond to questions point by point.
>
> *1. Did the authors experiment with other powerful interaction mechanisms?*
>
> We attempt to improve the MLP-based design, but it consistently struggled to perform effectively when the codebook size became large. Therefore, we decide to abandon further extensions of the MLP architecture and instead adopt a stronger ViT-based structure.
>
> *2. Why do the authors believe the ViT block (Patchify -> Process -> Unpatchify) architecture is so much more effective than a deep MLP for this task?*
>
> Compared to an MLP, the ViT is a well-validated and powerful module, integrating attention, ResNet blocks, effective normalization, and FFNs, while also scaling more effectively.
>
> The patchify–unpatchify design provides two main advantages:
>
> - a. It significantly reduces computational complexity under large codebook sizes (see experimental results in Table 10).
>
> - b. The initial patchify allows VQBridge to perform information interaction at a global level, followed by unpatchify to facilitate local interactions.
>
> This design helps better align the distributions of $P _ z$ and $\hat{C} _ z$.
>
> ---
> **Finally, we sincerely thank you for your valuable feedback and constructive discussions, which help us improve the paper. Please do not hesitate to reach out with any further questions. We remain fully available throughout the rebuttal period and look forward to your continued feedback.**

---

> > ### Comment · Reviewer_wFm6 · 2025-11-25
> >
> > Thank you for addressing my concerns. I will keep my score.

---

> > > ### Author Response · Authors · 2025-11-26
> > > **Thanks for the Quick Response!**
> > >
> > > We’re glad to hear that your concerns have been adequately addressed. We sincerely appreciate your professional and constructive feedback, which has helped make our work more solid and clear. If you have any further questions, please feel free to comment here. Thank you.

---

### Official Review · Reviewer_kziK · 2025-11-01

**Soundness:** 3
**Presentation:** 3
**Contribution:** 3
**Rating:** 4
**Confidence:** 3

**Summary:**

This paper addresses three key errors in Vector-Quantized Networks (VQNs): the bias introduced by the straight-through estimator (STE), the misalignment caused by one-step-behind updates, and the inefficiency resulting from sparse codebook utilization. These issues become particularly severe when scaling up codebook size and vector dimensions. The authors propose FVQ, a training framework that combines a novel projector module called VQBridge with learning rate annealing. VQBridge is designed as a "compress-process-recover" module utilizing ViT blocks to enable global joint optimization of the entire codebook. The experimental results are compelling: the method achieves 100% codebook utilization across various scales, including a massive 262k codebook, and establishes new state-of-the-art reconstruction performance. Furthermore, when integrated into autoregressive image generation pipelines, it significantly enhances generation quality, surpassing strong baselines such as VAR and DiT, thereby unlocking greater potential. The proposed framework also maintains high scalability across other methods.

**Strengths:**

1. The analysis is thorough and precise: This paper presents a clear and well-structured examination of the fundamental optimization challenges in VQ training. The explanations are well-supported by comprehensive comparative experiments and mathematical proofs. It is particularly noteworthy how the paper elaborates on the positive role of learning rate annealing in mitigating the one-step-behind update error, which significantly enhances the theoretical credibility.

2. High-Impact Results: The achievement of 100% codebook utilization at scale (up to 262k codes) is a major result . The reported reconstruction (rFID) and generation (FID) metrics are state-of-the-art, convincingly demonstrating the practical benefit of solving the codebook collapse problem.
3. The design is elegant and practical: The VQBridge module presents a clever and well-designed solution. The "compress-process-recover" pipeline not only makes scaling to large codebooks computationally feasible but also significantly enhances the representational capacity of the projection function, effectively overcoming the codebook collapse issue inherent in linear projectors. A key advantage is that the module is employed exclusively during training, introducing zero overhead during inference, which makes it highly appealing for practical deployment.
4. Extensive Experimentation: The authors perform a wide array of experiments, including ablation studies on the VQBridge components, scalability studies across codebook size, vector channels, and training epochs, and generalization tests on multi-code VQ methods (RQ-VAE, VAR). This leaves little doubt about the robustness and effectiveness of the proposed method.

**Weaknesses:**

1. Issues with the Mathematical Formulation of the Framework: Mathematical definitions are provided for the framework in the methodology section; however, there appears to be an apparent contradiction between the statement "For each group, the vectors are compressed into a single vector" and its corresponding matrix representation. Additionally, there seem to be inconsistencies in the tensor shapes described in the subsequent reshaping operation. The authors need to verify the accuracy of this part of the mathematical representation.

2. Computational Cost of Training: Although the inference cost remains unchanged, the introduction of the ViT-based VQBridge significantly increases the computational cost of training, particularly for large codebooks. The authors do not discuss these newly introduced training costs in the paper.
3. Ablation on Annealing Schedule: The paper heavily relies on the combination of VQBridge + Learning Annealing. While an ablation in Table 9 shows that the schedule type (cosine/linear) has a negligible impact, it does not ablate the *necessity* of the specific annealing parameters (warmup proportion, decay schedule) used. The robustness of the method to different annealing strategies could be better explored.
4. Comparison to Closest Contemporaries: While comparisons to VQGAN-LC are provided, a more direct and detailed comparison with other recent methods aiming for high usage with large codebooks (like IBQ) in terms of training efficiency, stability, and final performance on a level playing field would further strengthen the paper.

**Questions:**

1. Q1: Mathematical Formulation: Could the authors please verify the mathematical formulation in the methodology section? Specifically, what is the precise meaning behind the dimensionality of the matrices described in the framework?

2. Q2: Training Efficiency: Could the authors provide an estimate of the relative increase in total training time (e.g., for the 16k codebook configuration) when incorporating VQBridge compared to the VQGAN baseline? This would help practitioners better assess the trade-off between final performance and computational budget.
3. Loss Function: The authors mention in the paper that beyond estimation error, "codebook usage can therefore serve as a complementary metric." However, the paper does not explicitly state whether the framework involves a specific loss function for the codebook. Could the authors clarify how the codebook loss function is incorporated?

4. Q3: Design Choices: The scaling law between codebook size `K` and patch size `p` (`p = sqrt(K)/4`) is found empirically. Was there any intuition or theoretical justification for this relationship? Why does compressing the codebook by this specific factor lead to optimal performance?
5. Q4: Failure Mode Analysis: In the rare case that codebook usage does not reach 100%, what are the observed failure modes? Does the training become unstable, or does performance just plateau at a sub-optimal level?

---

> ### Author Response · Authors · 2025-11-20
> **Rebuttal by Authors**
>
> Dear Reviewer kziK,
>
> We thank you for valuable comments and questions, which help us improve our work. Our responses are below.
>
> ---
>
> > **[W1,Q1] Verify the mathematical formulation in the methodology section.**
>
> [A1] We sincerely thank the reviewer for detailed comments and for pointing out the issues in the formulas in our work.  We provide the corrected formula below and will carefully review the entire manuscript to identify and correct any other potential mistakes in the revision.
>
> $$
> \mathbf{h} _ g = \mathrm{LN}\big( \mathbf{C} _ g W _ {\text{comp}} \big), \quad g = 1, \dots, p.  \tag{8}
> $$
>
> $\mathbf{C} _ g\in \mathbb{R}^{1\times \frac{Kd}{p}}$, $W _ {\text{comp}} \in \mathbb{R}^{\frac{p}{Kd} \times d'}$, $\mathbf{h} _ g \in \mathbb{R}^{1\times d'}$ , $\mathbf{H}\in \mathbb{R}^{p \times d'}$ and $\mathbf{H}=[\mathbf{h} _ 1, \mathbf{h} _ 2,...,\mathbf{h} _ p]$.
>
> $K$ is the number of code vectors, $p$ is the number of groups, $d$ is the dimension of each code vector, and $d'$ is the dimension of the latent representation.
> $$
> \mathbf{\tilde{H}} = \mathrm{LN}(\mathbf{H}') W _ {\text{exp}}, \tag{10}
> $$
>
> $W _ {\text{exp}}\in \mathbb{R}^{d'\times \frac{Kd}{p}}$ and $\mathbf{\tilde{H}}\in \mathbb{R}^{p\times \frac{Kd}{p}}$.
>
>
>
>
>
> > **[W2,Q2] Computational Cost of Training**
>
> [A2] Thank you for your suggestions to provide more details on computational cost.
> First, we provide the computational cost of the VQBridge module (measured in FLOPs) in Appendix H, Table 10.
>
> Additionally, we include more details regarding the training resources below. It is important to note that the extra 3M parameters introduced by VQBridge account for only about 4% of the total parameters, which is a very small increase. Moreover, the overall training time remains almost unchanged (considering normal fluctuations in server throughput).
>
> | **Model** | **Module** | **Total Parameters** | **experimental Parameters** | **Training Time** |
> |------|----------|----------------|-------|---------------|
> | VQGAN | w/ nothing | 76.07M | 0M | ~27h |
> | FVQ | w/ linear | 76.14M | 0.66M | ~27h |
> | FVQ | w/ VQBridge | 79.23M | 3.15M | ~27h |
>
> > **[W3] Ablation on Annealing Schedule**
>
> [A3] We appreciate your detailed comments.
> The annealing schedule mainly addresses **Challenge 2: One-step-behind Update**. By reducing the learning rate while maintaining 100% codebook usage, we achieve the intended effect. While the specific warmup proportion and decay schedule may slightly affect overall training and cause minor variations in the final results, they do not affect our main conclusions.
>
> To address the reviewer’s concerns, we conduct experiments with different warmup proportions. The warmup proportion has only a minor effect on the results.
>
> | **Warmup Epoch** | **rFID** |
> |------|--------|
> | 2 |  1.32 |
> | 4 | 1.30  |
> | 6 | 1.35  |
>
> > **[W4] Comparison to Closest Contemporaries**
>
> [A4] Thank you for your suggestion to provide a more comprehensive comparison. Under comparable settings, we train all models for 40 epochs with a codebook dimension of 256. When the codebook size is 262k, IBQ reaches only 84% usage, and when the codebook size is 16k, IBQ reaches 96% usage. In contrast, our FVQ consistently achieves 100% usage across all settings, and as shown in Tables 1 and 2, our reconstruction and generation results also surpass those of IBQ, demonstrating the clear advantage of our approach.
>
> > **[Q3] Loss Function**
>
> [A5] We do not introduce any additional losses on VQGAN; the only change is that the codebook interacts through the proposed VQBridge. To keep the theoretical analysis clear and concise, we did not explicitly list all loss terms in the main text. The complete loss formulations are provided in Appendix F. Among these terms, $L_{\text{commit}}$ directly affects the codebook, as $z_q$ consists of the selected vectors from the codebook. We also visualize this loss in Appendix C, Figure 7.
>
> It is important to clarify that codebook usage is not a loss term. It is purely a monitoring statistic that reflects how fully the codebook is utilized during training, and it does not participate in optimization.
>
> **To be continued in next reply.**

---

> > ### Author Response · Authors · 2025-11-20
> > **Rebuttal by Authors (continued)**
> >
> > > **[Q4] Design Choices**
> >
> > [A6] Regarding design choices, there is no strict relationship such as $p = \sqrt{K}/4$. We adopt a 1D patchify operation with patch sizes of 4, 16, and 64, inspired by the 2D patchify settings in DiT ($2^2$, $4^2$, $8^2$).
> >
> > Our current experimental results indicate that increasing both the codebook size and the patch size simultaneously leads to noticeable performance improvements. This helps balance the computational load between the patchify stage and the ViT blocks, enabling the model to achieve better results.
> >
> > > **[Q5] Failure Mode Analysis**
> >
> > [A7] Across all VQBridge configurations, the codebook usage consistently reached 100%, and no failure cases were observed. During the training of Multi-Code Representation VQ, there are minor fluctuations in the early stages; however, the usage quickly stabilized and ultimately reached 100%.
> >
> > ---
> > **Thank you again for your insightful and professional comment, which helped make our work more complete and solid! If you have any further questions, please feel free to let us know. If you feel that all questions have been addressed, we would greatly appreciate it if you could kindly consider re-rating our work. Looking forward to hearing from you !**

---

> ### Comment · Reviewer_kziK · 2025-11-25
>
> Thank you for the detailed clarifications. My concerns have been largely addressed, and I will raise my rating accordingly.

---

> ### Author Response · Authors · 2025-11-26
> **Thanks for Your Kind Update!**
>
> We’re very happy to know our responses were helpful. Thank you so much for raising your rating. We really appreciate it and your thoughtful comments. Feel free to reach out anytime if anything else arises :-)!

---

### Official Review · Reviewer_M7kB · 2025-11-01

**Soundness:** 3
**Presentation:** 4
**Contribution:** 3
**Rating:** 8
**Confidence:** 3

**Summary:**

This paper studies presents a well justified and supported study into vector quantization for image tokenization. Generating discrete tokens from images is a non-trivial, yet crucial task for application in a myriad of modern vision methods. To address the training and scaling of the codebooks as compared to other methods, the author tackle challenges of Straight Through Estimation bias, lagging updates, and sparse updates. They present a method in VQBridge that uses mixing between entires at training, better initial alignment, and a different update method to handle the mentioned challenges. The authors accompany this introduced method and theory with good ablation studies and experiments to motivate the application of the introduced method in different studies, making this a very useful study in the explored topic.

**Strengths:**

1. The presented challenges are each sufficiently well motivated with examples and theory
2. Each presented challenge is suitably addressed in the building of the suggested VQBridge method
3. The suggested method is studied in a large domain of suitable of applications, showing good results in the strong suite of experimentation
4. Comprehensive ablation studies are included across different vector and codebook sizes, as well as the training efficiency is explained - which justify the claims of scalability
5. Generalization studies are added to show applicability in multi code representation as well
6. The effects of the use of the tokens produced are also studied in downstream tasks, which motivates the applicability of the method well

**Weaknesses:**

1. The argument for the proposed method could be made stronger with the inclusion of more details on computational cost. In particular, the added computational cost for training and how the training cost differs between the compared methods since only training epochs is used as an example
2. The larger codebooks retain the 100% utilization and are still shown to have great performance. Still, given that the patchification process induces sharing between entires, some insight into the uniqueness of the tokens generated and/or the distribution of active tokens could further ground the presented results.

**Questions:**

1. Expanding on the results in table 3, Is codebook size the only factor that influences the optimal patch size? Have the authors studied, as an example, how the patch size and vector channels interact ?
2. The smaller codebooks produced by FVQ seem to be provide strong results in reconstruction. Does the same benefit of not needing a larger codebook extend to downstream tasks ? Have the authors been able to study downstream tasks from models using different-size codebooks generated by FVQ ?

---

> ### Author Response · Authors · 2025-11-20
> **Rebuttal by Authors**
>
> Dear Reviewer M7kB,
>
> We thank you for valuable comments and questions, which help us improve our work. Our responses are below.
>
> ---
> > **[W1] The argument for the proposed method could be made stronger with the inclusion of more details on computational cost...**
>
> [A1] Thank you for your suggestions to provide more details on computational cost.
> First, we provide the computational cost of the VQBridge module (measured in FLOPs) in Appendix H, Table 10.
>
> Additionally, we include more details regarding the training resources below. It is important to note that the extra 3M parameters introduced by VQBridge account for only about 4% of the total parameters, which is a very small increase. Moreover, the overall training time remains almost unchanged (considering normal fluctuations in server throughput).
> We have added this in revision (see Appenxid J Traning Cost).
>
> | **Model** | **Module** | **Total Parameters** | **Extra Parameters** | **Training Time** |
> |------|----------|----------------|-------|---------------|
> | VQGAN | w/ nothing | 76.07M | 0M | ~27h |
> | FVQ | w/ linear | 76.14M | 0.66M | ~27h |
> | FVQ | w/ VQBridge | 79.23M | 3.15M | ~27h |
>
> > **[W2] ...some insight into the uniqueness of the tokens generated and/or the distribution of active tokens...**
>
> [A2] This is a very insightful and thought-provoking question.
>
> First, The patchification process does not affect the uniqueness of tokens. In vector quantization, the codebook selection is solely determined by the distance between the latent representation $z_e$ and the codebook entries $z_q$. If duplicated items existed in the codebook, some entries would be repeatedly selected, preventing the usage from reaching 100%. However, our observed usage is 100%, indicating that no duplicated entries are present.
>
> Second, regarding the token distribution, the t-SNE visualization in Figure 2 shows that the distributions of $z_e$ and $z_q$ under VQBridge are tightly aligned, without any isolated or drifting $z_q$ entries. This further confirms the effective utilization of the entire codebook and supports the observed 100% usage.
>
> > **[Q1] Expanding on the results in table 3, Is codebook size the only factor that influences the optimal patch size? Have the authors studied, as an example, how the patch size and vector channels interact ?**
>
> [A3] This is a highly insightful question.
>
> We have observed that the codebook size and patch size interact, as they jointly determine how the computational burden is distributed between the patchify stage and the ViT blocks.
>
> For further exploration, we conduct additional experiments varying both the **patch size** and the **vector channels**. The results show no noticeable interaction effects or performance degradation between these factors.
>
> | **Patch Size** | **Vector Channel** = 128 | **Vector Channel** = 256 |
> |------------|--------------------|--------------------|
> | 4          | 1.51               | 1.60               |
> | 16         | 1.32               | 1.31               |
> | 64         | 1.39               | 1.50               |
>
> > **[Q2] The smaller codebooks produced by FVQ seem to be provide strong results in reconstruction. Does the same benefit of not needing a larger codebook extend to downstream tasks ? Have the authors been able to study downstream tasks from models using different-size codebooks generated by FVQ ?**
>
> [A4] Thank you for this professional question. We evaluate FVQ models with different codebook sizes (4k, 16k, 65k) using LlamaGen-L as the generator. The results are summarized below:
>
> |  **Codebook Size**| 4k|16k| 65k |
> |------------|--------|---------|---|
> | **FID**    | 2.65 |2.39  | 2.30|
>
> As shown, increasing the codebook size consistently improves reconstruction and generation quality, consistent with observations from prior work such as IBQ[1]. However, a larger codebook also increases the parameter count of the generator’s final linear layer, which can make optimization more challenging. Simply adding a final linear layer may easily lead to model imbalance, and enlarging the overall model might be more effective. In general, larger codebooks require a larger-capacity generator to fully realize their benefits.
>
> Due to our limited training resources, we are currently unable to evaluate even larger codebook sizes with larger generators. We have included this limitation in the discussion in the revision.
>
> **References:**
>
> [1] Shi F, Luo Z, Ge Y, et al. Scalable image tokenization with index backpropagation quantization[C]//Proceedings of the IEEE/CVF International Conference on Computer Vision. 2025: 16037-16046.
>
> ---
> **We sincerely thank Reviewer M7kB for your thorough, insightful, and valuable feedback. We have made our best efforts to address each of your concerns and hope our responses clarify them. Please feel free to reach out with any further questions! We will be actively engaged in the discussion throughout the rebuttal period and look forward to your feedback!**

---

### Author Response · Authors · 2025-11-20
**Overall Author Rebuttal**

We sincerely thank the reviewers and chairs for their time and constructive feedback. We greatly appreciate the recognition of our work’s novelty, motivation, technical contributions, strong empirical performance, and scaling properties.

Here, we respond to one common question about

>**Why can a MLP projector still collapse under learning-rate annealing, while VQBridge remains stable?**

In our implementation, the baseline MLP projector is indeed **shared across all code vectors**: every code $\mathbf{c} _ k$ is mapped via the same function $f_\theta$, so that
$$
\hat{\mathbf{c}} _ k = f_\theta(\mathbf{c}_k), \quad k = 1,\dots,K.
$$
Thus, in a purely parametric sense, gradients from active codes do update the shared weights $\theta$ and thereby affect the outputs of *all* codes. The key question is why this still leads to codebook collapse, whereas VQBridge does not.

**1. Pointwise MLP + STE ⇒ low-rank, usage-sparse updates**

With the straight-through estimator, at each step **only the selected codes** participate in the loss. Denote the active set at iteration $t$ by $A_t$ and let $\mathbf{g} _ k = \partial L / \partial \hat{\mathbf{c}} _ k$. The gradient w.r.t. the shared MLP parameters is
$$
\frac{\partial L}{\partial \theta}
= \sum_{k \in A_t} J(\mathbf{c}_k)^\top \mathbf{g}_k,
$$
where $J(\mathbf{c} _ k) = \partial f _ \theta(\mathbf{c}_k)/\partial \theta$ is the Jacobian at input $\mathbf{c}_k$. When the codebook is large, $|A _ t| \ll K$, so $\partial L / \partial \theta$ lies in the low-dimensional subspace spanned by $\{J(\mathbf{c} _ k)\} _ {k \in A_t}$. Codes that are never selected contribute **zero** to this gradient.

An arbitrary code (including inactive ones) is updated indirectly via
$$
\Delta \hat{\mathbf{c}} _ k
\approx J(\mathbf{c} _ k)\Delta\theta _ t
= -\eta _ t J(\mathbf{c} _ k)\\sum_{i \in A_t} J(\mathbf{c}_i)^\top \mathbf{g}_i.
$$
Although this shows that active-code gradients do mathematically propagate to all codes, for an **inactive** code $k\notin A_t$ this effect is:
(i) weak in magnitude (since cross-Jacobian terms $J(\mathbf{c}_k)J(\mathbf{c}_i)^\top$ are typically small in high dimensions), and
(ii) not aligned with “where this code should go” in data space (it is driven entirely by gradients of other codes).

**2. Why these indirect updates are not enough once learning rate is annealed**

Codebook usage is determined by the nearest-neighbor assignment
$$
a(\mathbf{z};\theta) = \arg\min _ k \||\mathbf{z} - \hat{\mathbf{c}} _ k(\theta)\||^2,\quad
p _ k(\theta) = \Pr _ {\mathbf{z}}[a(\mathbf{z};\theta)=k].
$$
This assignment is a **piecewise constant** function of $\{\hat{\mathbf{c}}_k\}$: usage $p_k$ only changes when a code crosses a Voronoi boundary and becomes the closest vector for some encoder outputs $\mathbf{z}$.

In the early phase with a relatively large learning rate, the shared MLP benefits from stochastic optimization: large steps and noisy gradients allow the parameters to explore different regions of the code space, so a moderate level of usage can be achieved. However, once learning-rate annealing starts, the update
$$
\Delta\theta_t = -\eta_t \sum_{i\in A_t} J(\mathbf{c}_i)^\top \mathbf{g}_i, \quad \eta_t \downarrow 0,
$$
shrinks in magnitude and becomes dominated by the current active set. For an inactive code $k\notin A_t$, the induced movement
$$
\Delta \hat{\mathbf{c}}_k \approx J(\mathbf{c}_k) \Delta\theta_t
$$
is then **both very small** (due to the decayed $\eta_t$ and weak cross-Jacobian terms) and **not guided by its own data assignments**. In practice, this results in tiny, noise-like drifts that are insufficient to cross any Voronoi boundary. Consequently, its nearest-neighbor assignments do not change, and its usage $p_k$ remains essentially zero.

In this sense, such codes are “effectively frozen” **in terms of assignment and usage**, even though the shared MLP parameters continue to be updated. The optimization settles into a local minimum where only a small subset of codes is active, and overall codebook usage collapses, as observed for the MLP baseline under annealing.

**To be continued in next reply.**

---

> ### Author Response · Authors · 2025-11-20
> **Overall Author Rebuttal (continued)**
>
> **3. Why VQBridge behaves differently**
>
> VQBridge is designed to avoid precisely this failure mode. Instead of mapping each code independently, VQBridge first **patchifies** the codebook $C \in \mathbb{R}^{K \times d}$, applies several ViT blocks with global self-attention, and then **unpatchifies** to obtain $\hat{C} = f _ \theta(C)$.
> Through self-attention, each output code depends explicitly on a **group of input codes**, and the gradients are distributed via dense attention weights $\alpha _ {kj}$. In gradient form, this yields
> $$
> \frac{\partial L}{\partial \theta}
> \propto \sum _ {k,j} \alpha _ {kj} J _ {kj}^\top \mathbf{g} _ j,
> $$
> so strong gradients from frequently used codes are **directly propagated** to many other codes in the same patch (and across patches via multi-head attention). This produces a **densely coupled** update over the entire codebook rather than a low-rank update dominated by the current active subset.
>
> As a result, even under learning-rate annealing, previously inactive codes receive meaningful, data-aligned updates that can move them into regions where they become nearest neighbors for some encoder outputs. This prevents them from being “usage-frozen” and allows VQBridge to maintain ~100% codebook usage across different codebook sizes and with annealing, in contrast to the shared MLP projector.
>
> In summary, the collapse of the shared MLP projector under annealing is not caused by a lack of parameter sharing, but by its **pointwise structure combined with STE**, which yields low-rank, usage-sparse updates that are too weak and misaligned to revive inactive codes once the learning rate is decayed. VQBridge introduces explicit cross-code interaction, resulting in strong, dense gradient coupling that keeps the entire codebook trainable and avoids collapse.
>
> ---
> For non-common questions, we've responded below the review comment of each reviewer.
>
> Best,
>
> The Authors

---

### Author Response · Authors · 2025-12-01
**Final Summary by Authors**

Dear PCs, SACs, ACs,

We sincerely  appreciate thorough reviews and valuable feedback. These comments and discussions have greatly helped us improve our work, and we have addressed most of the concerns.

We appreciate the recognition of our paper’s strengths:
- Our work provides a rigorous and well-structured analysis of fundamental VQ optimization challenges, supported by mathematical proofs, comparative experiments, and clear explanations.
- It also delivers high-impact empirical results, including 100% codebook utilization at scale and state-of-the-art rFID and FID performance. These results are enabled by the elegant and practical VQBridge design, which introduces zero inference overhead and is thoroughly validated through extensive ablations, scalability studies, and generalization experiments.
- Clear motivation, well-structured presentation, and intuitive illustrations make the methodology easy to understand and follow.

We have also addressed most of the concerns:
- Provided additional results comparing training overhead, showing that VQBridge introduces minimal extra cost.
- Added further discussion on why VQBridge outperforms MLPs, along with additional experiments and model design details.
- Included more experimental results across different codebook sizes for generation tasks, as well as additional ablation studies.

The rebuttal version PDF manuscript is also updated accordingly.

Four out of five reviewers have responded and confirmed that their concerns have been addressed, without raising any additional issues. In particular, reviewer #kziK noted in their response, “I will raise my rating accordingly,” and had indicated a weak accept (from rating 4 to rating 6).

Finally, we thank all reviewers for recognizing our contribution. We believe our work advances the community by rigorously analyzing VQ optimization challenges, introducing the practical and scalable VQBridge module to mitigate codebook collapse, and demonstrating state-of-the-art performance across multiple tasks and codebook sizes. We will release code, data, and checkpoints to facilitate further research.

We sincerely thank you for your time and careful consideration.

Best regards,

The Authors

---

### Meta-Review · Area_Chair_h76Y · 2026-01-07

**Summary:**

Reviewers praised the rigorous analysis of VQ optimization challenges, the elegant VQBridge design achieving 100% codebook utilization at scale (up to 262k), SOTA reconstruction, and strong downstream impact (vanilla AR outperforming VAR by 0.5 rFID and DiT by 0.2 rFID). Minor concerns included training overhead, theoretical depth on why VQBridge succeeds where MLP fails, novelty, and additional ablations/comparisons.

**Reviewer Concerns:**

Most concerns were effectively addressed in the rebuttal: training overhead shown minimal (~same wall-clock time, +4% params), detailed explanation of MLP collapse vs. VQBridge stability (with math), added ablations (annealing, patch/channel interaction), extra comparisons (IBQ, more datasets/baselines), and generation results across codebook sizes. No major concerns remain outstanding.

**Reviewer Scores:**

- Reviewer M7kB (initial 8): Likely remains 8 (strong positives, minor questions addressed).
- Reviewer kziK (initial 4): Raised to 6 as stated in discussion.
- Reviewer wFm6 (initial 6): Maintained 6.
- Reviewer 93aA (initial 6): Maintained 6.
- Reviewer pncl (initial 6): Maintained 6.

---

### Decision · Program_Chairs · 2026-01-26

Accept (Poster)